# Evaluation of oceanic and atmospheric trajectory schemes in the TRACMASS trajectory model v6.0

Kristofer Döös[1], Bror Jönsson[2], and Joakim Kjellsson[3]

[1]Department of Meteorology, Stockholm University, SE-10691 Stockholm, Sweden.
[2]Department of Geosciences, Princeton University, Guyot Hall, Princeton, NJ 08544, USA
[3]Atmospheric, Oceanic, and Planetary Physics, University of Oxford, UK

*Correspondence to:* Kristofer Döös (doos@misu.su.se)

**Abstract.** Three different trajectory schemes for oceanic and atmospheric general circulation models are compared in two different experiments. The theories of the trajectory schemes are presented showing the differential equations they solve and why they are mass conserving. One scheme assumes that the velocity fields are stationary for set intervals of time between saved model outputs and solves the trajectory path from a differential equation only as a function of space, i.e. "stepwise stationary". The second scheme is a special case of the "stepwise-stationary" scheme, where velocities are assumed constant between GCM outputs, it uses hence a "fixed GCM time step". The third scheme uses a continuous linear interpolation of the fields in time and solves the trajectory path from a differential equation as a function of both space and time, i.e. "time-dependent". The trajectory schemes are tested "off-line", i.e. using the already integrated and stored velocity fields from a GCM. The first comparison of the schemes uses trajectories calculated using the velocity fields from a high resolution ocean general circulation model in the Agulhas region. The second comparison uses trajectories calculated using the wind fields from an atmospheric reanalysis. The study shows that using the "time-dependent" scheme over the "stepwise-stationary" scheme greatly improves accuracy with only a small increase in computational time. It is also found that with decreasing time steps the "stepwise-stationary" scheme becomes increasingly more accurate but at increased computational cost. The "time-dependent" scheme is therefore preferred over the "stepwise-stationary" scheme. However, when averaging over large ensembles of trajectories the two schemes are comparable, as intrinsic variability dominates over numerical errors. The "fixed GCM time step" scheme is found to be less accurate than the "stepwise-stationary" scheme, even when considering averages over large ensembles.

## 1 Introduction

The Lagrangian view of the ocean and atmospheric circulation describes fluid pathways and the connectivity of different regions, which are not readily obtained from an Eulerian perspective. Lagrangian studies often require trajectory calculations using some algorithm that transforms the Eulerian velocity fields, e.g. winds or currents, into trajectories. Although observed velocities can be used, it is much more common to use velocities simulated by a General Circulation Model (GCM). The purpose of this work is to test the different schemes used in the TRACMASS trajectory model (version 6.0) here named the "fixed GCM time step" (Blanke and Raynaud, 1997; Döös, 1995), "stepwise stationary" (Döös et al., 2013) and "time-

dependent" (de Vries and Döös, 2001). These schemes have previously only been tested using highly idealised velocity fields. Here we will test them velocity fields simulated by comprehensive GCMs for both the ocean and atmosphere.

The TRACMASS trajectory model (Jönsson et al., 2015) has been continuously updated through the years since it was first introduced by Döös (1995). Version 6.0 represents the latest version, which includes the ability to run TRACMASS with the "time-dependent" scheme by de Vries and Döös (2001) on GCM fields. TRACMASS now also supports many different types of vertical coordinates used in atmosphere and ocean GCMs. The code has also been made more structured and user friendly.

The original feature of TRACMASS and the related Ariane model (Blanke and Raynaud, 1997) is that they solve the trajectory path through each model grid cell with an analytical solution of a differential equation, which depends on the velocities on the faces of the model grid box. This is different from iterative schemes such as the commonly used 4th-order Runge-Kutta (RK4). The TRACMASS schemes have many advantages, e.g. mass conservation within the grid cell in the same way as the GCM itself, as well as fast trajectory computation. Furthermore, as the solution to the differential equation is unique, trajectories can be calculated forward in time and subsequently backward in time to arrive at exactly the original position. This makes it possible to trace the origins of water or air masses as long as stochastic parameterisations (cf. Döös and Engqvist (2007)) are not activated.

The first trajectory scheme tested here, "fixed GCM time step", is strictly only valid for stationary velocity fields. It can, however, be used with time-varying velocity fields by dividing the time between GCM outputs into intermediate steps and assuming velocities are stationary during the step. The velocities in an intermediate step are found by linear interpolation between two GCM outputs and hence named "stepwise stationary". However, using intermediate steps increases the computational cost. The "time-dependent" scheme does not assume that the fields are stationary and uses instead continuous bilinear interpolation both in space and time.

The fact that the "stepwise-stationary" scheme uses stepwise-stationary velocities is logical when the scheme is used "online", i.e. integrated into a GCM and thus having the same time step as the GCM itself. When the scheme is used "off-line", i.e. separately from the GCM and after the velocity fields have been stored, the time step is the time between two GCM outputs, which typically is a much longer period than the GCM time step. As the "stepwise-stationary" scheme assumes that velocities are constant during the time step of the trajectory scheme, processes faster than the GCM output frequency are lost.

An alternative to the "stepwise-stationary" scheme was introduced by de Vries and Döös (2001), where the trajectory solution was not only solved analytically in space as was done by Blanke and Raynaud (1997), as well as Döös (1995), but also analytically in time between the GCM outputs. This leads to a more complex differential equation to be solved and integrated as the trajectory progresses through space and time (Döös et al., 2013). The advantage of this "time-dependent" scheme by de Vries and Döös (2001) is that it does not require any intermediate time steps between the model output times and can instead be integrated analytically between the GCM outputs. This method contrasts to the "fixed GCM time step" scheme by Blanke and Raynaud (1997) and the "stepwise stationary" by Döös et al. (2013) as well as schemes such as Euler forward or RK4 methods (Butcher, 2008; Fabbroni, 2009), where the trajectories are integrated forward in time with as short time steps as possible. A comprehensive review of different trajectory codes as well as the fundamental kinematic framework behind these can be found in van Sebille (2016).

In section 2 we describe the three different trajectory schemes and how they are integrated in time in both Ocean General Circulation Models (OGCMs) and Atmospheric General Circulation Models (AGCMs). In section 3 we test the three trajectory schemes with two different velocity fields, one from an OGCM and one from an AGCM, using various statistics. This study is concluded in section 4 with a summary and discussion of the main results of the trajectory schemes and their tests.

## 2 Trajectory scheme theory

The trajectory schemes used in TRACMASS are all mass conserving but make different assumptions regarding the time evolution of the Eulerian velocity and pressure fields. The schemes rely on the assumption that, within a grid cell the three velocities components are only linear functions of their corresponding directions, i.e. $u = u(x)$, $v = v(y)$ and $w = w(z)$. An alternative approach is to assume that $u = u(x,y,z)$, $v = v(x,y,z)$ and $w = w(x,y,z)$ inside a grid cell, which might be more realistic
in terms of representing unresolved motions. However, no such information is generally provided by GCMs. Furthermore, it would also require that the mass transports through the grid faces are unchanged in order to satisfy the continuity equation of the GCM.

The trajectory schemes integrate the trajectories from the volume or mass transports through the grid-box faces in contrast to many other trajectory schemes that only use the velocity fields. We will first describe how these fluxes are computed and then
the three different trajectory schemes.

### 2.1 Mass and volume flux

The TRACMASS trajectory schemes are mass conserving as they, like the GCM, deal with the transport across the grid faces and the transport is only interpolated linearly between the two opposite faces in a grid box. The trajectories will hence never cross a grid boundary.
A GCM mesh is generally spherical or curvilinear. The longitudinal ($\Delta x_{i,j}$) and the latitudinal ($\Delta y_{i,j}$) grid lengths will hence be functions of their horizontal positions $i,j$ on a curvilinear grid. The vertical coordinate in a GCM has a depth level thickness $\Delta z_{i,j,k}^n$, where $k$ is vertical level, and $n$ is time step. Note that the vertical resolution can vary not only vertically but also both horizontally and in time, which makes it possible to use any vertical coordinate e.g. sigma (Marsh and Megann, 2002), z-star, pressure or hybrid coordinates (Kjellsson and Döös, 2012b). The horizontal mass transports through the eastern
and northern faces, respectively, of the $i,j,k$ grid box at time step $n$ are given by

$$U_{i,j,k}^n = \rho_{i,j,k}^n u_{i,j,k}^n \Delta y_{i,j} \Delta z_{i,j,k}^n, \tag{1}$$
$$V_{i,j,k}^n = \rho_{i,j,k}^n v_{i,j,k}^n \Delta x_{i,j} \Delta z_{i,j,k}^n. \tag{2}$$

The zonal velocity $u_{i,j,k}^n$ and the meridional velocity $v_{i,j,k}^n$ are in the above equations on a C-grid. It is, however, possible to use the velocities from A and B-grid models, where the velocities are instead at the corners of the grid cell, leading to

30 $$U_{i,j,k}^n = \rho_{i,j,k}^n \tfrac{1}{2} \left( u_{i,j,k}^n + u_{i,j-1,k}^n \right) \Delta y_{i,j} \Delta z_{i,j,k}^n, \tag{3}$$
$$V_{i,j,k}^n = \rho_{i,j,k}^n \tfrac{1}{2} \left( v_{i,j,k}^n + v_{i-1,j,k}^n \right) \Delta x_{i,j} \Delta z_{i,j,k}^n. \tag{4}$$

This averaging of two horizontal grid points in order to have the perpendicular velocity to the grid box in the middle on the grid face is exactly how a B-grid model discretises the equations, when e.g. solving the continuity equation.

Note that the mass transport can be replaced by the volume transport in models that assume the fluid to be incompressible, which is the case for most OGCMs. In other models (most AGCMs), we may use the hydrostatic approximation to write

$$\Delta p_{i,j,k}^n = \rho_{i,j,k}^n g \Delta z_{i,j,k}^n. \tag{5}$$

where $g$ is gravity and $p$ is air pressure. The mass transports through the lateral grid faces in the AGCM expressed by Eqs. (1, 2) will use Eq. (5) to determine $\Delta z$ and hence become

$$U_{i,j,k}^n = u_{i,j,k}^n \Delta y_{i,j} \Delta p_{i,j,k}^n / g \tag{6}$$

$$V_{i,j,k}^n = v_{i,j,k}^n \Delta x_{i,j} \Delta p_{i,j,k}^n / g. \tag{7}$$

The vertical mass transport can similarly be computed from the vertical velocity $w_{i,j,k}$ through the upper face of the grid box so that

$$W_{i,j,k}^n = \rho_{i,j,k} w_{i,j,k}^n \Delta x_{i,j} \Delta y_{i,j}. \tag{8}$$

The vertical velocity would in the equation above be taken directly from the stored velocity fields from the GCM. It is, however, in order to guarantee mass conservation, advantageous to instead calculate the vertical transport $W_{i,j,k}^n$ from the continuity

equation as the TRACMASS trajectory schemes rely on mass or volume continuity.

The continuity equation, which expresses conservation of mass, states that

$$\frac{\partial \rho}{\partial t} + \frac{\partial (\rho u)}{\partial x} + \frac{\partial (\rho v)}{\partial y} + \frac{\partial (\rho w)}{\partial z} = 0. \tag{9}$$

Integrating Eq. (9) over a finite grid box of volume $\Delta x \Delta y \Delta z$ we obtain

$$\frac{\partial M_{i,j,k}}{\partial t} + U_{i,j,k} - U_{i-1,j,k} + V_{i,j,k} - V_{i,j-1,k} + W_{i,j,k} - W_{i,j,k-1} = 0, \tag{10}$$

where $M_{i,j,k}$ is the mass of the grid box. The rate of mass change of the grid box $\partial M_{i,j,k}/\partial t$ can on the other hand be due to 1) compression in an compressible GCM and/or to 2) grid-box volume change, which generally in a GCM is due to the time dependence of the vertical resolution so that the thickness of model layers vary in time.

The mass of the grid box is

$$M_{i,j,k}^n = \rho_{i,j,k}^n \Delta x_{i,j} \Delta y_{i,j} \Delta z_{i,j,k}^n, \tag{11}$$

where $n$ is the time level of the stored GCM fields so that time is $t = n\Delta t_G$ and $\Delta t_G$ is the time interval between two stored GCM fields.

The vertical mass transport through the top of the grid box is obtained by discretising Eq. (10) between two stored time levels:

$$W_{i,j,k}^n = W_{i,j,k-1}^n - \left[ U_{i,j,k}^n - U_{i-1,j,k}^n + V_{i,j,k}^n - V_{i,j-1,k}^n + \frac{(\rho_{i,j,k}^n \Delta z_{i,j,k}^n - \rho_{i,j,k}^{n-1} \Delta z_{i,j,k}^{n-1})}{\Delta t_G} \Delta x_{i,j} \Delta y_{i,j} \right], \tag{12}$$

which is computed by integration from the bottom and upwards with the bottom boundary condition $W_{i,j,0} = 0$. This is the same way the vertical velocity is computed in the GCM except that we use the stored horizontal velocities and the grid-size thicknesses to ensure that they satisfy the time dependency correctly.

In many OGCM, the fluid is considered to be incompressible and thus the density is constant and $\rho$ can be dropped from all equations in order to have volume flux instead of mass flux in the calculations. The vertical volume transport through the top of the grid box becomes

$$W_{i,j,k}^n = W_{i,j,k-1}^n - \left[ U_{i,j,k}^n - U_{i-1,j,k}^n + V_{i,j,k}^n - V_{i,j-1,k}^n + \frac{(\Delta z_{i,j,k}^n - \Delta z_{i,j,k}^{n-1})}{\Delta t_G} \Delta x_{i,j} \Delta y_{i,j} \right]. \tag{13}$$

If, additionally, the vertical resolution is time independent, the last term can be neglected and thus

$$W_{i,j,k}^n = W_{i,j,k-1}^n - \left( U_{i,j,k}^n - U_{i-1,j,k}^n + V_{i,j,k}^n - V_{i,j-1,k}^n \right). \tag{14}$$

On the other hand, in many AGCMs there is both compressibility of the air and a time dependence of the vertical resolution, which is generally expressed in pressure and hence

$$W_{i,j,k}^n = W_{i,j,k-1}^n - \left[ U_{i,j,k}^n - U_{i-1,j,k}^n + V_{i,j,k}^n - V_{i,j-1,k}^n + \frac{(\Delta p_{i,j,k}^n - \Delta p_{i,j,k}^{n-1})}{g \Delta t_G} \Delta x_{i,j} \Delta y_{i,j} \right], \tag{15}$$

where Eq. 5 has been used. Note that in the case of "off-line" calculations, one may instead use centred or forward finite time differences in Eqs. 12, 13 and 15.

## 2.2 The stationary case

This scheme assumes that the velocity and pressure fields are in steady state and was introduced by Blanke and Raynaud (1997) and used and developed for ocean mass transport studies by Döös (1995). The velocity inside a grid cell is found by assuming that it is only a function of its direction, i.e. $u = u(x), v = v(y), w = w(z)$. Linear interpolation gives the zonal velocity

$$u(x) = u_{i-1,j,k} + \frac{x - x_{i-1,j}}{\Delta x_{i,j}} \left( u_{i,j,k} - u_{i-1,j,k} \right), \tag{16}$$

and similarly for $v(y)$ and $w(z)$. Note that the calculation of the vertical mass transport $W_{i,j,k}^n$ through the top face of a grid box, with the Eqs. 12 – 15, only involves the velocities on the considered grid box. A 3D dependency of the velocities ($u = u(x,y,z)$, $v = v(x,y,z)$ and $w = w(x,y,z)$) would require velocities from other grid boxes, which could potentially break the mass conservation of Eqs. 12 – 15.

To calculate the zonal position, $x$, of a trajectory, we use $u = dx/dt$, and write Eq. 16 as the differential equation

$$\frac{dx}{dt} - \frac{u_{i,j,k} - u_{i-1,j,k}}{\Delta x_{i,j}} x + \frac{x_{i-1,j}}{\Delta x_{i,j}} \left( u_{i,j,k} - u_{i-1,j,k} \right) - u_{i-1,j,k} = 0.$$

The drawback by solving the above differential equation is that $\Delta x$ is not constant and a horizontal grid face is rarely rectangular in a GCM. The solution will hence depend on the position of the trajectory in each grid box. Döös (1995) used therefore a $\Delta x$ corresponding to the average latitudinal position of the trajectory in each grid box, which was obtained by computing the

trajectories several times in each grid box. Blanke and Raynaud (1997) made this unnecessary by non-dimensionalising the position and used volume fluxes instead of velocities. By substituting $x$ for a non-dimensional position $r \equiv x/\Delta x_{i,j}$ and $t$ for a scaled time $s \equiv t/(\Delta x_{i,j} \Delta y_{i,j} \Delta z_{i,j,k})$, we get

$$\frac{dr}{ds} + \beta r + \delta = 0, \tag{17}$$

where $F = dr/ds$ is the zonal volume or mass flux, and $\beta \equiv F_{i-1,j,k} - F_{i,j,k}$ and $\delta \equiv -F_{i-1,j,k} - \beta r_{i-1}$ are constants. Its solution describes the zonal displacement within the grid box between the faces and is found using the initial condition $r(s_0) = r_0$ of its zonal position so that

$$r(s) = \left( r_0 + \frac{\delta}{\beta} \right) e^{-\beta(s-s_0)} - \frac{\delta}{\beta}. \tag{18}$$

The scaled time $s_1$ becomes

$$s_1 = s_0 - \frac{1}{\beta} \log \left[ \frac{r_1 + \delta/\beta}{r_0 + \delta/\beta} \right], \tag{19}$$

where $r_1 = r(s_1)$ is given by either $r_{i-1}$ or $r_i$, when a trajectory enters the western or eastern grid face, respectively. The logarithmic factor in Eq. (19) can be expressed as $\log[F(r_1)/F(r_0)]$.

For a trajectory reaching the grid face $r = r_i$ or $r = r_{i-1}$ both $F(r_1)$ and $F(r_0)$ must be of the same sign in order for Eq. (19) to have a solution. If $F(r_1)$ and $F(r_0)$ are of opposite signs there is a zero zonal transport at a position between $r_1$ and $r_0$

and this position is reached exponentially slow.

The above procedure is repeated for meridional and vertical displacements, where now $r = y/\Delta y_{i,j}$ or $r = z/\Delta z_{i,j,k}$. This yields non-dimensional position, $r_1$, and scaled time, $s_1$, for the zonal, meridional and vertical displacements of the trajectory, respectively, inside the grid box under consideration. The smallest transit time $s_1 - s_0$ and the corresponding $r_1$ denote through which grid face of the grid box the trajectory will exit and move into the adjacent one. The exact displacements in the other

two directions are then computed using the smallest $s_1$ in the corresponding Eq. (18).

Note that Eqs. (18)-(19) are not valid if the transport fields across the grid box are constant, i.e. when $(F_{i-1,j,k} = F_{i,j,k})$, since it would imply a division by zero with $\beta = 0$ in both equations. The differential equation then simplifies to

$$\frac{dr}{ds} + \delta = 0, \tag{20}$$

which has the solution

$$r(s) = -\delta(s - s_0) + r_0, \tag{21}$$

and the scaled time $s_1$ is

$$s_1 = s_0 - \frac{r_1 - r_0}{\delta}. \tag{22}$$

If $F_{i-1,j,k} = F_{i,j,k}$, TRACMASS instead uses Eq. 21, 22.

## 2.3 "Stepwise-stationary" and "Fixed GCM time step" integrations

The trajectory scheme above is, strictly speaking, only valid for stationary fields. The scheme is, however, possible to use for time-dependent fields by assuming that the velocity and surface-elevation fields are stationary during a limited time interval. The "stepwise-stationary" method presented here consists of assuming that the fields are stationary during intermediate time steps between two GCM outputs and then updated successively as new fields become available. If this is undertaken "on-line", i.e., in the same time as the GCM is integrated, this time interval will simply be the same as the time step the GCM is integrated, which is typically between minutes to a few hours in a global GCM. If instead the trajectories are calculated "off-line", the time interval between GCM fields will be at least as often as the fields have been stored by the GCM, at intervals that can be days or even months.

A linear time interpolation of the velocity fields between two GCM velocity fields permits a simple way to have shorter time steps by which the fields are updated in time. The time interval between two GCM velocity fields is $\Delta t_G$ and the shorter time interval at which the fields are interpolated is $\Delta t_i$ as illustrated by Fig. 1. The number of intermediate time steps is hence the ratio $I_S = \Delta t_G/\Delta t_i$. For any quantity in the GCM output, $F$, the value at intermediate time step $m$, located between GCM outputs $n-1$ and $n$, is

$$F(t^m) \equiv F^m = \frac{t^m - t^{n-1}}{\Delta t_G}(F^n - F^{n-1}) + F^{n-1}. \tag{23}$$

The coefficients $\beta, \delta$ in Eq. (17) are updated when a trajectory moves from one grid box to another. Thus, the time step for the trajectory, i.e. $s_1 - s_0$, may be shorter than the intermediate time step, $\Delta t$. $\Delta t_i$ is hence the maximum possible time step for a given $I_S$, but is often shorter if the spatial grid spacing ($\Delta x$, $\Delta y$, $\Delta z$) is small and $\Delta t_G$ long. We will therefore test TRACMASS by imposing constant velocities for the entire $\Delta t_G$ in order to mimic other codes such as the Ariane code based on Blanke and Raynaud (1997), which do not make any temporal interpolations of the velocity fields. This particular case of the "stepwise-stationary" scheme with constant velocity fields for the entire period between two GCM outputs will be denoted the "fixed GCM time step". These two schemes together with a truly time dependent scheme, described in next section, will be tested.

## 2.4 Analytical time integration with the "time-dependent" scheme

The "stepwise-stationary" integration method presented in the previous section assumes that the velocity and the grid box thicknesses remain constant throughout the time step, and only spatial variations of velocity are accounted for. Another approach is to interpolate the velocity fields, not only in space within the grid box, but also in time between the GCM outputs. This approach, introduced in TRACMASS by de Vries and Döös (2001), is more accurate but involves a more advanced differential equation to be solved and integrated along the trajectories. Accounting for both spatial and temporal variations of velocities in the trajectory scheme render intermediate time steps unnecessary. We will later show that using a large number of intermediate

steps, the "stepwise-stationary" scheme approaches this "time-dependent" scheme asymptotically.

The "time-dependent" scheme can be derived in the same way as Eq. 17, but instead of a linear interpolation in space, we use a bilinear interpolation in both space and time. As before, we use non-dimensional position $r = x/\Delta x$, and scaled time $s \equiv t/(\Delta x \Delta y \Delta z)$, where the denominator is the volume of the particular grid box. For a zonal volume or mass flux $F$ a bilinear interpolation in space and time yields

$$
\begin{aligned}
F(r,s) \;=\; & F_{i-1}^{n-1} + (r - r_{i-1})(F_i^{n-1} - F_{i-1}^{n-1}) + \\
& + \frac{s - s^{n-1}}{\Delta s}\left[ F_{i-1}^n - F_{i-1}^{n-1} + (r - r_{i-1})(F_i^n - F_{i-1}^n - F_i^{n-1} + F_{i-1}^{n-1}) \right],
\end{aligned}
\tag{24}
$$

$\Delta s$ is the scaled time step between two data sets:

$$
\Delta s = s^n - s^{n-1} = (t^n - t^{n-1})/(\Delta x \Delta y \Delta z) = \Delta t_G/(\Delta x \Delta y \Delta z),
\tag{25}
$$

where $\Delta t_G$ is the time step between two data sets in true time dimension (seconds). The vertical grid box spacing is for models with time dependent grid cell thicknesses replaced with an average between the two time steps: $\left(\Delta z^n + \Delta z^{n-1}\right)/2$. Similar expressions for the meridional and vertical directions can be derived.

Connecting the local transport to the time derivative of the position with $F = dr/ds$, the following differential equation is obtained:

$$
\frac{dr}{ds} + \alpha\,r\,s + \beta\,r + \gamma\,s + \delta = 0,
\tag{26}
$$

where the coefficients are defined by

$$
\begin{aligned}
\alpha &\equiv -\frac{1}{\Delta s}\left(F_i^n - F_{i-1}^n - F_i^{n-1} + F_{i-1}^{n-1}\right), \tag{27}\\
\beta &\equiv F_{i-1}^{n-1} - F_i^{n-1} - \alpha\,s^{n-1}, \tag{28}\\
\gamma &\equiv -\frac{1}{\Delta s}\left(F_{i-1}^n - F_{i-1}^{n-1}\right) - \alpha\,r_{i-1}, \tag{29}\\
\delta &\equiv -F_{i-1}^{n-1} + r_{i-1}(F_i^{n-1} - F_{i-1}^{n-1}) - \gamma\,s^{n-1}. \tag{30}
\end{aligned}
$$

Different analytical solutions exist for the three cases: $\alpha > 0$, $\alpha < 0$ and $\alpha = 0$, which together cover all possible values of $\alpha$. The acceleration, inside the $r - s$ grid box, is $d^2r/ds^2 = -\alpha r - \gamma$, which is constrained by a linear $r$-dependent term proportional to $\alpha$ and the constant $\gamma$.

### 2.4.1 The case $\alpha > 0$

For this case, we define the time-like variable $\xi = (\beta + \alpha\,s)/\sqrt{2\alpha}$ and get

$$
r(s) = \left(r_0 + \frac{\gamma}{\alpha}\right)e^{\xi_0^2 - \xi^2} - \frac{\gamma}{\alpha} + \frac{\beta\gamma - \alpha\delta}{\alpha}\sqrt{\frac{2}{\alpha}}\left[D(\xi) - e^{\xi_0^2 - \xi^2}D(\xi_0)\right],
\tag{31}
$$

where Dawson's integral

$$D(\xi) \equiv e^{-\xi^2} \int_0^{\xi} e^{x^2} dx \tag{32}$$

has been used, as well as, the initial condition $r(s_0) = r_0$. An example of trajectories in this case is illustrated in Fig. 2a, with given values of $F_{i-1}^{n-1}$, $F_i^{n-1}$, $F_i^n$ and $F_{i-1}^n$. We see here that $\alpha > 0$ occurs when the flow changes from divergence in the i-direction at time step $n-1$ to convergence at time step $n$.

### 2.4.2 The case $\alpha < 0$

When $\alpha < 0$, $\xi$ becomes imaginary. By defining $\zeta \equiv i\xi = (\beta + \alpha s)/\sqrt{-2\alpha}$, Eq. (31) can be re-expressed as

$$r(s) = \left( r_0 + \frac{\gamma}{\alpha} \right) e^{\zeta^2 - \zeta_0^2} - \frac{\gamma}{\alpha} - \frac{\beta\gamma - \alpha\delta}{\alpha} \sqrt{\frac{\pi}{-2\alpha}} e^{\zeta^2} \left[ \mathrm{erf}(\zeta) - \mathrm{erf}(\zeta_0) \right], \tag{33}$$

where the error function $\mathrm{erf}(\zeta) = (2/\sqrt{(\pi)} \int_0^{\zeta} e^{-x^2} dx$. An example of trajectories for this case is illustrated in Fig. 2b. We see here that $\alpha < 0$ occurs when the flow changes from convergence in the i-direction at time step $n-1$ to divergence at time step $n$.

### 2.4.3 The case $\alpha = 0$

The solution of Eq. (26) when $\alpha = 0$ is

$$r(s) = \left( r_0 + \frac{\delta}{\beta} \right) e^{-\beta(s-s_0)} - \frac{\delta}{\beta} + \frac{\gamma}{\beta^2} \left[ 1 - \beta s + (\beta s_0 - 1)e^{-\beta(s-s_0)} \right]. \tag{34}$$

This case would normally not occur in a realistic GCM integration, but if for some reason such as a chosen constant field in time or space, $\alpha$ will be zero, since $F_i^n - F_{i-1}^n = F_i^{n-1} + F_{i-1}^{n-1}$. Note that if the fields are in steady state, Eq. (34) is reduced to become identical to the stationary solution of Eq. (18). An example of trajectories in this stationary case is illustrated in Fig. 2c.

If instead $\alpha = 0$ since the fields are constant in space, i.e. the transport across the grid cell is constant ($F_i = F_{i-1}$), then we also have $\beta = 0$, which leads to a simplification of Eq. (26):

$$\frac{dr}{ds} + \gamma s + \delta = 0, \tag{35}$$

with the solution

$$r(s) = r_0 - \frac{\gamma}{2} \left( s^2 - s_0^2 \right) - \delta \left( s - s_0 \right). \tag{36}$$

An example of trajectories in this case with constant fields in space is illustrated in Fig. 2d.

## 2.5 The transit time

A major difference between the "time-dependent" and the "stepwise-stationary" schemes is that in the time-dependent scheme, the transit times $s_1 - s_0$ cannot in general be obtained explicitly with the "time-dependent" scheme in contrast to the "stepwise-stationary" analytical solution of Eq. (18). Using the solutions given by Eqs. (31)–(34), the relevant root $s_1$ of

$$r(s_1) - r_1 = 0 \tag{37}$$

has to be computed numerically for each direction. We will now describe how the roots $s_1$ and the corresponding exiting grid face $r_1$ can be determined. The displacement of the trajectory inside the grid box under consideration then proceeds as previously discussed for stationary velocity fields.

We now determine the roots $s_1$ of Eq. (37) and the corresponding $r_1$ needed to calculate trajectories inside a grid box. In
10
what follows, $s^{n-1} \leqslant s_0 < s^n$ and the relevant roots $s_1$ are to be in the interval of $s_0 < s_1 \leqslant s^n$ . We also focus on the cases $\alpha > 0$ and $\alpha < 0$, since the forthcoming considerations can easily be adapted for the case of $\alpha = 0$. For numerical purposes, we use

$$\frac{\beta\gamma - \alpha\delta}{\alpha} = \frac{F_i^n F_{i-1}^{n-1} - F_i^{n-1} F_{i-1}^n}{F_i^n - F_{i-1}^n - F_i^{n-1} + F_{i-1}^{n-1}}, \tag{38}$$

$$\frac{\gamma}{\alpha} = \frac{F_{i-1}^n - F_{i-1}^{n-1}}{F_i^n - F_{i-1}^n - F_i^{n-1} + F_{i-1}^{n-1}} - r_{i-1}, \tag{39}$$

$$\xi = \frac{F_{i-1}^{n-1} - F_i^{n-1} + \alpha(s - s^{n-1})}{\sqrt{2\alpha}}, \tag{40}$$

$$\zeta = \frac{F_{i-1}^{n-1} - F_i^{n-1} + \alpha(s - s^{n-1})}{\sqrt{-2\alpha}}. \tag{41}$$

As above, $s$ is the scaled time. The coefficient in Eq. (38) appearing in Eqs. (31) and (33) is exactly zero when either the $r_{i-1}$ or $r_i$ grid face represents a solid boundary, so that transport $F_i$ or $F_{i-1}$ is zero for all $n$, respectively. In these instances, the opposite grid face fixes $r_1$ , and the root $s_1 > s_0$ can be computed analytically. If there is no solution, we take $s_1 = s^n$. When
20
all three transit times equal $s^n$, the trajectory will not move into an adjacent grid box but will remain inside the original one. Its new position is subsequently determined, and the next time interval is considered.

The roots of Eq. (37) have to be computed numerically if $(\beta\gamma - \alpha\delta)/\alpha \neq 0$. This is also true for locating the extrema of the solutions given by Eqs. (31) and (33). Alternatively, one can consider the case $F(r, s) = 0$ using Eq. (24) to analyse where possible extrema are located. It follows that in the $s$-$r$-plane, the extrema lie on a hyperbola of the form $r = (as + b)/(c + ds)$.
25
Obviously, only the parts defined by $s^{n-1} \leq s \leq s^n$ and $r_{i-1} \leq r \leq r_i$ are relevant. Depending on which parts of the hyperbola, if any, lie in this "box" and satisfy the initial condition $r(s_0) = r_0$, the trajectory $r(s)$ exhibits none, one, or at most two extrema. In the latter case, the trajectory will not cross either the grid face at $r_{i-1}$ or the one at $r_i$ (see Fig. 2 for an example). Hence, the trajectories $r(s)$ determining the transit time $s_1 - s_0$ will have at most one extremum, i.e., there is at most one change of sign in the local transport $F$.

An efficient way of proceeding is as follows: first consider the grid face at $r_i$. For a trajectory to reach this grid face, the local transport must be nonnegative, which depends on the signs of the transport $F_{i-1}^n$ and $F_i^n$. Four distinct configurations may arise between the model outputs ($s^{n-1} < s < s^n$), where the calculation of the trajectory takes place:

1. $F(r_i, s) > 0$ for $s^{n-1} < s < s^n$.

2. The sign of $F(r_i, s)$ changes from positive to negative at $s = s^*$, where $s^{n-1} < s^* < s^n$

3. The sign of $F(r_i, s)$ changes from negative to positive at $s = s^\#$, where $s^{n-1} < s^\# < s^n$.

4. $F(r_i, s) < 0$ for $s^{n-1} < s < s^n$.

These four cases are illustrated by the four panels of Fig. 3.

For case 1, we evaluate $r(s^n)$ using the appropriate analytical solution. If, in addition $r(s^n) \geq r_i$, then the trajectory has crossed the grid-box face $r = r_i$ at $s_1 \leq s^n$ as shown by the trajectories A, B and C in Fig. 3. If the initial transport $F(r_0, s_0) < 0$, the trajectory may have crossed the opposite grid face at an earlier time as illustrated by trajectory C in Fig. 3. This is only possible if case 3 applies for the grid face at $r_{i-1}$ and $s^\# > s_0$, in which case it is determined whether $r(s^\#) \leq r_{i-1}$. If this is not the case, there is a solution to $r(s_1) - r_1 = 0$ for $r_1 = r_i$ and $s_0 < s_1 \leq s^n$. Subsequently, this root can be calculated numerically using a root-solving algorithm (Press et al., 2007). But if $r(s^n) < r_i$ or, if applicable, $r(s^\#) \leq r_{i-1}$, we proceed by considering the other grid faces. The arguments for the grid face at $r_{i-1}$ are similar to those relating to $r_i$.

If case 2 applies and $s_0 < s^*$, we add here to the considerations given in case 1 using $s^*$ instead of $s^n$. If there is a root for $r_1 = r_i$, then $s_0 < s_1 \leq s^*$. This root is illustrated by trajectory D in Fig. 3 with $(r_1, s_1) = (r_i, s_{1D})$.

For case 3, we follow the procedure given by case 1. If there is a root for $r_1 = r_i$, then $s^\# < s_1 \leq s^n$. This root is illustrated by trajectory E in Fig. 3 with $(r_1, s_1) = (r_i, s_{1E})$.

For case 4, no solution of Eq. (37) is possible for $r_1 = r_i$, since all trajectories exit through the grid face located at $r_{i-1}$ as illustrated by trajectory G in Fig. 3 or will not reach any grid face during the time interval $s^{n-1} < s < s^n$. We must then instead search for an exit through another of the six grid faces.

All these considerations are applied to each of the three spatial directions in order to determine through which of the 6 grid faces the trajectory will exit and at which position on the corresponding grid face.

Since the trajectories are unique solutions to Eq. 26 and the continuity equation is respected, the TRACMASS trajectories will therefore never hit any solid boundary such as the coast or the sea floor unless the sedimentation option is activated, where an extra velocity is imposed, a feature that was introduced in TRACMASS by Corell and Döös (2013).

An example of the evolution of trajectories calculated with the three different schemes within a time-space cell for $\alpha > 0$ is shown in Fig. 4. The trajectories computed with the "stepwise-stationary" scheme approaches the trajectory computed with the "time-dependent" with increasing number of intermediate time steps ($I_S$). The "fixed GCM time step" trajectory can, however, not follow the "time-dependent" one since it does not update the velocities between the GCM outputs and consequently deviates immediately as it leaves the initial point $(r_0, s^{n-1})$.

## 3 Tests with different velocity fields

The results obtained from the "stepwise-stationary" scheme are now compared with those from the "time-dependent" trajectory schemes using two different sets of velocity fields. The first uses a high resolution OGCM with z-star coordinates. The second uses a global Atmospheric General Circulation Model with hybrid pressure coordinates. For the "stepwise-stationary" scheme, five different settings of $I_S$, i.e. the number of intermediate steps, are tested. The "fixed GCM time step" is also tested for comparison, although it is not a standard feature of TRACMASS.

### 3.1 Ocean trajectories with a high resolution OGCM

Oceanic velocity fields for this case were obtained from a simulation with the 3.6 version of the NEMO ocean model (Madec, 2016) in a global ORCA12 configuration. The horizontal resolution of the ORCA12 grid is approximately $1/12°$, corresponding to $\Delta x \approx 6$ km at $50°$ latitude. Model fields were available as 5-day averages every 5 days. The configuration uses 75 $z^*$ vertical levels with partial bottom cells, where $\Delta z$ ranges from $\sim 1$ m at the surface to $250$ m in the deepest parts of the ocean. The $z^*$ coordinate approach permits large-amplitude free-surface variations relative to the vertical resolution (Adcroft and Campin, 2004). In the $z^*$ formulation, the variation of the column thickness due to sea-surface undulations is not concentrated to the surface level, as in the z-coordinate formulation, but is equally distributed over the full water column. Thus the vertical levels naturally follow the sea-surface variations, which also implies that they are time dependent and we therefore have used Eq. (12) to calculate the vertical transport in TRACMASS with a time dependent $\Delta z^n$ in the equation. The model was forced with 6-hourly atmospheric fields from what is known as the Drakkar Forcing Set, version 4 (DFS4) (Brodeau et al., 2010). Sub-grid processes were represented using $125$ m$^2$ s$^{-1}$ Laplacian iso-neutral tracer diffusion, and $-1.25 \cdot 10^{10}$ m$^4$ s$^{-1}$ bi-Laplacian viscosity.

TRACMASS has been applied to this specific model integration already by Nilsson et al. (2013), where it was compared with surface drifters in the Agulhas region. This is also the region where we are going to test TRACMASS because of its complex time-dependent dynamics with travelling eddies, known as "Agulhas rings", which "leak" Indian-Ocean water into the Atlantic Ocean as part of the Conveyor Belt. 2193 trajectories were started, evenly spread over 4 grid horizontal boxes at all depths in the Indian Ocean, and followed for 50 days as shown in Figs. 5 and 6.

### 3.2 Atmospheric trajectories with an AGCM

In order to test the trajectory schemes in the atmosphere we have used the ERA-Interim reanalysis (Dee et al., 2011) from the European Centre for Medium-range Weather Forecasts (ECMWF) simulated with the IFS (Integrated Forecasting System) model. In this ERA-Interim data set, the vertical coordinate is a terrain-following hybrid coordinate (Simmons and Burridge, 1981), where the pressure at the lower interface of level $k$ is given by $p_k = A_k + B_k p_s$, where $p_s$ is the surface pressure and $A_k$ and $B_k$ are parameters at the level $k \in [0, 60]$, with $p_{60} = p_s$ and $p_0 = 0$. As in the NEMO ocean model, the grid cell thickness varies in time, and we calculate vertical mass flux from the continuity equation (Eq. 15). The ERA-Interim data used here had a horizontal resolution of $1.25°$ and is available 6-hourly ($\Delta t_G$). Trajectories are shown in Fig. 7. They were initiated

every 6 hours from a grid cell air column over Eyjafjallajökull Volcano eruption during 14-18 March 2010. The trajectories were evenly distributed horizontally and started in exactly same positions for the tests with different time steps using the "stepwise-stationary" scheme and "time-dependent" case.

### 3.3 Lagrangian statistics

The average distance between the trajectories obtained with the "time-dependent" scheme and the five different "stepwise-stationary" cases as well as the "fixed GCM time step" case are shown in Fig. 8. The distances from the "time-dependent" trajectories after 50 days for the OGCM case and after 10 days for the AGCM case are presented in Table 1. These average distances have been possible to compute since of all the individual trajectories were started in the exact same positions for the different cases. Results clearly show that the distance between trajectories calculated with the "stepwise-stationary" scheme

and those calculated with the "time-dependent" scheme decreased as the number of intermediate time steps were increased. The "fixed GCM time step" case, i.e. when no intermediate time steps are used, shows the greatest distance to the "time-dependent" case.

     Standard Lagrangian statistics have also been computed for the ocean trajectories (Fig. 9), with the definitions given in the

Appendix. The *relative* and *absolute dispersion* as well as the *mean displacement* of the trajectory cluster show how the cluster will disperse and move in time. They reveal a similar pattern, where only the "fixed GCM time step" case differs from the others. The "fixed GCM time step" differs already after 3 to 4 days, which should be related to the fact that the GCM velocities are updated every 5 days ($= \Delta t_G$) in this OGCM case.

     The *Lagrangian velocity autocorrelation*, which describes the correlation of the velocity of the trajectories at one time with

that of previous times, shows in Fig. 9 how all cases except the "fixed GCM time step" give nearly the exact same correlation. The Lagrangian time scale, which is computed from the autocorrelation and is a measure of the memory of the trajectories, reflects the same feature with a Lagrangian time scale of approximately 3.9 days for the "time step" and the "time-dependent" cases but a slightly shorter time scale of 3.4 days for the "fixed GCM time step" case. The Lagrangian time scale based on observations with surface drifters is clearly shorter than this both for the Global Ocean (Döös et al., 2011) and in the Agulhas

region (Nilsson et al., 2013). This relatively shorter Lagrangian time scale (hence closer to observations for the "fixed GCM time step") is simply due to the abrupt changes in the velocity fields every time these are updated. A realistic shortening of the Lagrangian time scale can only be obtained by incorporating finer scales by increasing the GCM resolution or adding sub-grid parameterisations to the trajectories.

     The power spectra computed from the Lagrangian velocities show that the "fixed GCM time step" was more energetic than

the other schemes, which all yielded nearly identical results. This is the case for all frequencies. There is also a weak maximum at 4 cycles/day (6 hours), which remains unexplained, although it may be related to the fact that the OGCM uses 6-hourly atmospheric forcing.

### 3.4 Lagrangian stream function and residence time

The mass conservation properties of the used trajectory schemes make it possible to calculate mass transports between different sections in the model domain (Döös, 1995). The approach is that one can associate each trajectory particle with a mass or volume transport. This requires that enough trajectories are computed to fill the model grid in space and time with a sufficient number of trajectories. Lagrangian stream functions can be calculated by summing over trajectories representing a desired path (Blanke et al., 1999; Döös et al., 2008; Kjellsson and Döös, 2012b). The difference between the "Lagrangian" and the more common "Eulerian" stream functions is that with the Lagrangian one can isolate a particular path between a starting and an ending section in the ocean or the atmosphere.

The influence of the different trajectory schemes on the inter-ocean exchange of water masses, which takes place in the Agulhas region, has been evaluated by calculating Lagrangian stream functions. Fig. 10 shows the Lagrangian barotropic stream function computed from trajectories using the "time-dependent" scheme and the "fixed GCM time step" scheme. Lagrangian decomposition has been used to compute two separate stream functions for each scheme, one from trajectories entering the Atlantic and one from those returning back into the Indian Ocean via the Agulhas retroflection region. The "time-dependent" scheme favours slightly (one additional stream line) the entering into the Atlantic compared to the "fixed GCM time step" scheme. This is also clearly visible when computing the residence time, i.e. the time trajectories stay within the Agulhas region as shown in the lower righthand panel of Fig. 9. The first particles start to exit the Agulhas region as defined by the map in Fig 10 after 50 days. The number of trajectory particles then decays exponentially with an e-folding time of about 210 days. This is rather similar for all trajectory-scheme integrations. There is, however a clear difference in the results where the trajectories exit. The "fixed GCM time step" scheme results in 38 % flowing into the Atlantic and 59 % into the Indian Ocean after 800 days. All the other trajectory integrations yield very similar results but with 46 % flowing into the Atlantic and 52 % into the Indian Ocean. This suggests that the "fixed GCM time step" scheme does not capture the same behaviour as the other schemes.

We have repeated the above ocean-trajectory experiment by releasing the particles in other time periods and increasing the ensemble size. The results only changed marginally.

### 3.5 Computational speed

In addition to the higher accuracy of the "time-dependent" scheme, it was also shown to be computationally faster than the "stepwise-stationary" scheme with intermediate time steps. In order to quantify this difference we compared the computational time for the different schemes using analytical velocity fields describing inertia oscillations (Döös et al., 2013), where no data needed to be read nor written since the velocity fields have a known analytical solution and disk storage was switched off. These computational times are shown in the last column of Table 1, which have been normalised by dividing with the time obtained with the "time-dependent" scheme. The "stepwise-stationary" scheme was only as computationally fast as the "time-dependent" scheme when no extra intermediate time steps were taken between two readings of the velocity fields ($I_S = 1$) or when using "fixed GCM time steps". When the number of intermediate time steps was increased to $12,000$, the "stepwise-stationary"

scheme was more than 1000 times slower. 12,000 intermediate steps was also approximately the number of intermediate time steps required in order to obtain as accurate results as those obtained from the "time-dependent" scheme.

## 4    Discussion and Conclusions

The two trajectory schemes available in TRACMASS have here been inter-compared by calculating Lagrangian statistics,
transports and the distances between the trajectories. This has been done for both oceanic and atmospheric applications. The "stepwise-stationary" scheme assumed that the velocity fields were stationary for the duration of a user defined intermediate time step between model output fields. These velocities are, however updated with a linear interpolation in time when crossing a model grid face. The "time-dependent" scheme does not assume that the velocity is in steady state during any time interval since it solves the differential equations of the trajectory path not only in space but also in time. This continuous evolution of
the "time-dependent" scheme makes it more accurate than the "stepwise-stationary" scheme without any significant increase in computational expense.

In addition to these two TRACMASS schemes, we have tested a "fixed GCM time step" scheme, which is in fact a special case of the "stepwise-stationary" scheme but with velocity fields always remaining in steady state until a new GCM data set is reloaded in order to mimic the Ariane trajectory model (Blanke and Raynaud, 1997). A consequence of only updating the fields
at the GCM output times is that the velocities are assumed to be in steady state for long periods and then changed abruptly with a discontinuity.

The accuracy of the schemes has been evaluated by comparing the distance between particles that have been started from the same positions but with different trajectory schemes, and how this distance evolves in time. This distance was shown to depend on the scheme and the number of intermediate time steps for the "stepwise-stationary" case. The average distance as a
function of time between the trajectories obtained from the different schemes are shown in Fig. 8 as well as their end position distances in Table 1.

The study has shown that the TRACMASS "time-dependent" scheme is likely to be more accurate as well as faster than the "stepwise-stationary" scheme with intermediate steps. It remains to be shown how the trajectory schemes used in the present study compare to other trajectory schemes, such as e.g. Runge-Kutta, which could be used where mass conservation is not
important.

The "stepwise-stationary" scheme needed up to 12,000 intermediate time steps to give as accurate trajectory paths as the "time-dependent" scheme, which is more than a thousand times as computationally expensive when reading and writing is excluded. The distance between trajectories calculated with the "time-dependent" scheme and those obtained with the "stepwise-stationary" scheme decreased as the number of intermediate time steps is increased. The greatest distance was obtained when
no temporal variations between GCM outputs at all were considered, i.e. with the "fixed GCM time step" scheme. We thus conclude that the "time-dependent" scheme is the most accurate of those tested here for two reasons. Firstly for theoretical reasons since the "time-dependent" scheme does not assume stationary velocities during any period of time. Secondly the trajectories computed with the "stepwise-stationary" scheme converge towards those computed with the "time-dependent" scheme

for increasing numer of intermediate time steps. A future study could be to calculate trajectories first using fields stored at each GCM time step and second using fields stored at longer time intervals. In the first case, trajectories would be very accurate and could represent a "truth", and the second case could be used to evaluate which scheme is the closest to the "truth".

The Lagrangian statistics such as relative and absolute dispersion as well as Lagrangian velocity autocorrelation functions and power spectra showed almost identical results for the "time-dependent" and the "stepwise-stationary" schemes. The "fixed GCM time step" showed, however, some differences from the other two schemes. E.g. the dispersion after 3-4 days was slightly larger for using a "fixed GCM time step", which might be explained by an abrupt change every time the GCM velocities are updated compared to the smoother transition of the two other schemes. The results show that the "fixed GCM time step" method does not capture the same behaviour of trajectories as the other schemes. The Lagrangian statistics are also clearly affected by the model resolution and the time sampling of the GCM fields (Döös et al., 2011; Kjellsson and Döös, 2012a; Kjellsson et al., 2013; Nilsson et al., 2013). Future improvements to the TRACMASS model will involve improvements of the sub-grid turbulence parameterisations, which could give more realistic dispersion properties.

The mass conservation of the trajectory schemes in the present study arises from that 1) mass transports across the grid faces are used in the same way as in the GCM itself instead of velocities as in most other trajectory schemes, 2) the mass transport is linearly interpolated within the grid box, where there is otherwise no information of the velocity from the GCM and that this enables us to set up a differential equation, which has an analytical solution of the trajectory within the grid box. The different trajectory schemes, although mass conserving, will not yield the same results in terms of transports between different sections. The mass transport was tested in the Agulhas experiment, where the "fixed GCM time step" scheme favoured relatively the Agulhas retroflection with more trajectories returning into the Indian compared to the "time-dependent" and "stepwise-stationary" schemes. This difference in mass transport can be explained by the delicate path of the Agulhas leakage, which requires an accurate temporal evolution so that particles can be retained in Agulhas rings. This was better achieved by the "time-dependent" and "stepwise-stationary" schemes than by the "fixed GCM time step" scheme.

The TRACMASS trajectory code with corresponding schemes has been improved and become more accurate and user friendly over the years. An outcome of the present study is that we strongly recommend the use of the "time-dependent" scheme based on de Vries and Döös (2001) in favour of the "stepwise-stationary" scheme. We would also like to dissuade the use of the more primitive "fixed GCM time step" scheme, which is used in other trajectory codes since the velocity fields remain stationary for longer periods creating abrupt discontinuities in the velocity fields, and yielding inaccurate solutions. We have here only tested one OGCM and one AGCM simulation, but we speculate that at coarser resolution in both space and time, the differences obtained with the two schemes would increase. However, in non eddying simulations (e.g. 1° ocean models) this may not be true due to the low variability of the flow.

The TRACMASS strict requirement of mass conservation makes it, however, necessary to have complete velocity fields on the original GCM grid in order to use mass or volume transports in and out of each model grid box. This requirement of mass conservation will always be somewhat more demanding than for other trajectory codes, since it requires a total understanding of the various GCM coordinate systems as well as incorporating them in the TRACMASS framework. This state of affairs is in marked contrast to what holds true for various trajectory codes that only require velocity fields with no mass conservation.

## 5 Code availability

TRACMASS version 6.0 is freely available for research purposes at https://github.com/TRACMASS. In addition, the code is archived at http://dx.doi.org/10.5281/zenodo.34157.

## Appendix A: Lagrangian-statistics definitions

The Lagrangian statistics used in the present work (shown in Figs. 8 and 9) are here defined. See e.g. LaCasce (2008) for a detailed derivation.

The *average distance* between the different trajectory calculations as presented in Fig. 8 is defined as

$$D_B^2(t) \equiv \frac{1}{M-1} \sum_{m=1}^{N} \sum_{i=1}^{2} (x_{i,m}(t) - \hat{x}_{i,m}(t))^2. \tag{A1}$$

It is hence the distance between the two trajectories $x_{i,m}(t)$ and $\hat{x}_{i,m}(t)$, where $t$ is the time, $M$ the total number of trajectories

of the cluster and $i$ the spatial coordinate index (*i.e.* the zonal, meridional or vertical position of the $m$-th trajectory $x_{i,m}(t)$). The two trajectories $x_{i,m}(t)$ and $\hat{x}_{i,m}(t)$ will have the same initial position $(x_{i,m}(t_0) = \hat{x}_{i,m}(t_0))$ but will then evolve differently since different trajectory schemes are used to compute their paths. In the present study, we only consider the horizontal dispersion. The vertical dispersion is, however, an important measure of the vertical mixing in the ocean but beyond the scope of the present study.

The mean position of the trajectory cluster is defined as

$$\overline{x_i(t)} \equiv \frac{1}{M} \sum_{m=1}^{M} x_{i,m}(t). \tag{A2}$$

The *relative dispersion* is defined as the mean-square displacement of the trajectories relative to the time-evolving mean position:

$$D_R^2(t) \equiv \frac{1}{M-1} \sum_{m=1}^{M} \sum_{i=1}^{2} \left( x_{i,m}(t) - \overline{x_i(t)} \right)^2. \tag{A3}$$

The *absolute dispersion* is defined in the same way, but relative to the initial position of the cluster:

$$D_A^2(t) \equiv \frac{1}{M-1} \sum_{m=1}^{M} \sum_{i=1}^{2} \left( x_{i,m}(t) - \overline{x_i(t_0)} \right)^2, \tag{A4}$$

where $t_0$ is the initial time of the trajectory.

The *mean displacement* is defined as the displacement from the origin as a function of time

$$D_D(t) \equiv \frac{1}{M} \sum_{m=1}^{M} \sqrt{\sum_{i=1}^{2} [x_{i,m}(t) - x_{i,m}(t_0)]^2}. \tag{A5}$$

The *Lagrangian velocity* is obtained by using a non-centered finite difference:

$$u_{i,m}(t^n) \equiv \frac{dx_{i,m}(t^n)}{dt} \approx \frac{x_{i,m}(t^n) - x_{i,m}(t^{n-1})}{t^n - t^{n-1}}, \tag{A6}$$

wihere $n$ is the time level. Similarly, the acceleration was calculated by finite differencing of the velocity:

$$a_{i,m}(t^n) \equiv \frac{du_{i,m}(t^n)}{dt} \approx \frac{u_{i,m}(t^n) - u_{i,m}(t^{n-1})}{t^n - t^{n-1}}. \tag{A7}$$

Note how velocity is not defined at the first position, and acceleration is not defined at the first velocity.

The *Lagrangian velocity autocorrelation* describes the correlation of the velocity at one time with that of previous times. The definition is

$$R(\tau) = \frac{\sigma^2(\tau)}{\sigma^2(\tau = 0)} \approx R(t^q)\frac{(\sigma(t^q))^2}{(\sigma(t^0))^2} \tag{A8}$$

where $\sigma^2(\tau)$ and $\sigma^2(\tau = 0)$ are the Lagrangian velocity auto-covariances for time lag $\tau$ and no lag, respectively. $q$ is the
discrete time step and $R_q$ is the autocorrelation at time step $q$. $\sigma^2(\tau)$ is defined as

$$\sigma^2(\tau) = \lim_{T\to\infty} \frac{1}{T}\int_0^T \mathbf{u}'(t+\tau)\cdot\mathbf{u}'(t)\,dt \approx (\sigma(t^q))^2 \equiv \sum_{i=1}^2 \frac{1}{N-q-1}\sum_{n=1}^{N-q-1} u_i'(t^n)u_i'(t^{n+q}), \tag{A9}$$

where $u_i'(t^n) = u_i(t^n) - \overline{u}_i$ and $\overline{u}_i$ is a time average of the segment. Note that the total velocity autocovariance is the sum of the zonal and meridional components, $\sigma^2 = \sigma^2_{i=1} + \sigma^2_{i=2}$.

The *Lagrangian time scale* is defined as

$$T_L = \int_0^\infty R(\tau)\,d\tau. \tag{A10}$$

This is a measure of the *memory* of a trajectory, i.e. the time lag during which the Lagrangian velocity is correlated. When computing this integral, the point where $R(\tau) = 0$ for the first time is used here as upper bound. This truncation is perhaps the most commonly used, due to the often noisy character of the auto-correlation function, $R(\tau)$ for large $\tau$.

*Acknowledgements.* The authors wish to thank Peter Lundberg for constructive comments. This work has been financially supported by the Bolin Centre for Climate Research and by the Swedish Research Council (Grant 2015-04442). Joakim Kjellsson is supported by the UK Natural Environment Research Council grant NE/K012150/1: "Poles apart: why has Antarctic sea ice increased, and why can't coupled climate models reproduce observations?". The GCM integrations and the trajectory computations were performed using resources provided by the Swedish National Infrastructure for Computing (SNIC) at the National Supercomputer Centre at Linköping University (NSC).

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

**Table 1.** The table shows the average distance between the "time-dependent" integrated trajectories and the "stepwise-stationary" integrated ones at the end of simulations, which is 50 days for the OGCM and 10 days for the AGCM. $I_S$ is the number of intermediate time steps between two GCM outputs. The "maximum time step" stands for the intermediate time step lengths ($\Delta t_i$), which are used in the different trajectory integrations. The last column is the computational time normalised with regard to the "time-dependent" case, where theoretical velocity fields are used to compute trajectories, i.e. with no data reading or writing.

| $I_S$ | Distance to "time dependent" | | $T_L$ | Maximum Time step | | Normalised |
| | OGCM | AGCM | AGCM | OGCM | OGCM | computational |
| | $[km]$ | $[km]$ | $[days]$ | $\Delta t_i$ | $\Delta t_i$ | time |
|---|---|---|---|---|---|---|
| "Fixed" | 769 | 4992 | 3.44 | $\equiv$5 d | $\equiv$6 h | 0.830 |
| 1 | 276 | 3835 | 3.88 | 5 d | 6 h | 0.830 |
| 12 | 242 | 2971 | 3.86 | 10 h | 30 min | 2.110 |
| 120 | 103 | 1752 | 3.86 | 1 h | 3 min | 14.03 |
| 1,200 | 28 | 1079 | 3.87 | 6 min | 18 s | 132.0 |
| 12,000 | 6 | 1002 | 3.87 | 36 s | 2 s | 1191 |
| Time dependent | 0 | 0 | 3.87 | 5 d | 6 h | 1.000 |

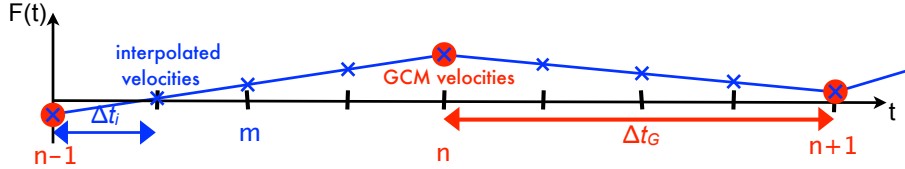

**Figure 1.** Schematic illustration of how the transport fields $F(t)$ are updated and interpolated in time between the stored GCM data, which are read in at the time $t^n$ and are separated in time by the time interval $\Delta t_G$ (in red). The fields are then linearly interpolated at the points in blue with intermediate time steps. The number of intermediate time steps between two GCM velocities is in this example $I_S = \Delta t_G / \Delta t_i = 4$.

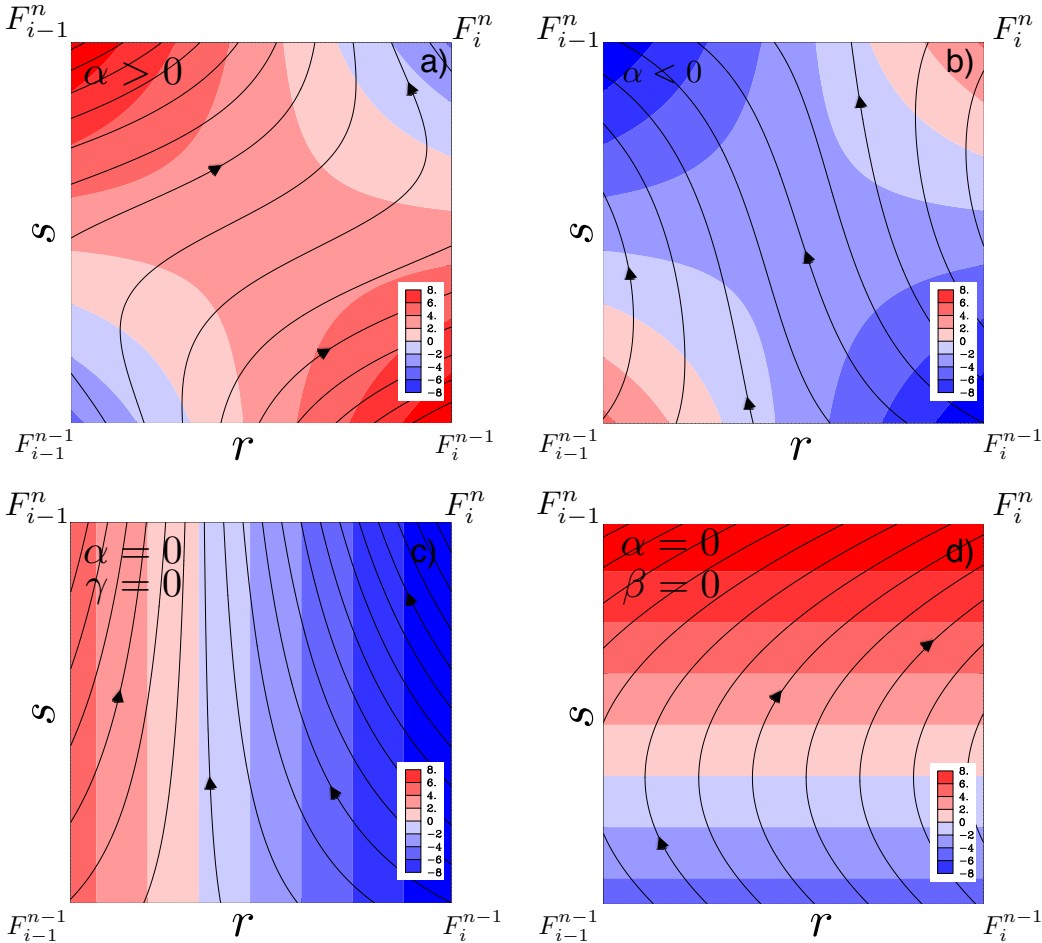

**Figure 2.** Examples of how trajectories calculated with the "time-dependent" scheme evolve as a function of the transport $F$ in the space interval $r_{i-1} < r < r_i$ and in the time interval $s^{n-1} < s < s^n$, which hence corresponds to an interval between two GCM outputs ($\Delta t_G$) and of a grid box ($\Delta x$, $\Delta y$ or $\Delta z$). The colour shows the transport values $F$ obtained by the bilinear interpolation between the four corners ($F_{i-1}^{n-1}$, $F_i^{n-1}$, $F_i^n$ and $F_{i-1}^n$). a) $\alpha > 0$ with two corners of transport in the negative direction ($F < 0$), which correspond to westward, southward or downward directions and one corner flowing in the opposite direction. b) $\alpha < 0$. c) $\alpha = 0$ and $\gamma = 0$ corresponds to the stationary fields, which results in an $F$ field that only changes in the ($r$) direction. d) $\alpha = 0$ and $\beta = 0$ corresponds to the constant fields in space but which vary in time. Note that the $F = 0$ line between the red and blue colours corresponds to static flow, which results in "vertical" trajectories in the figures.

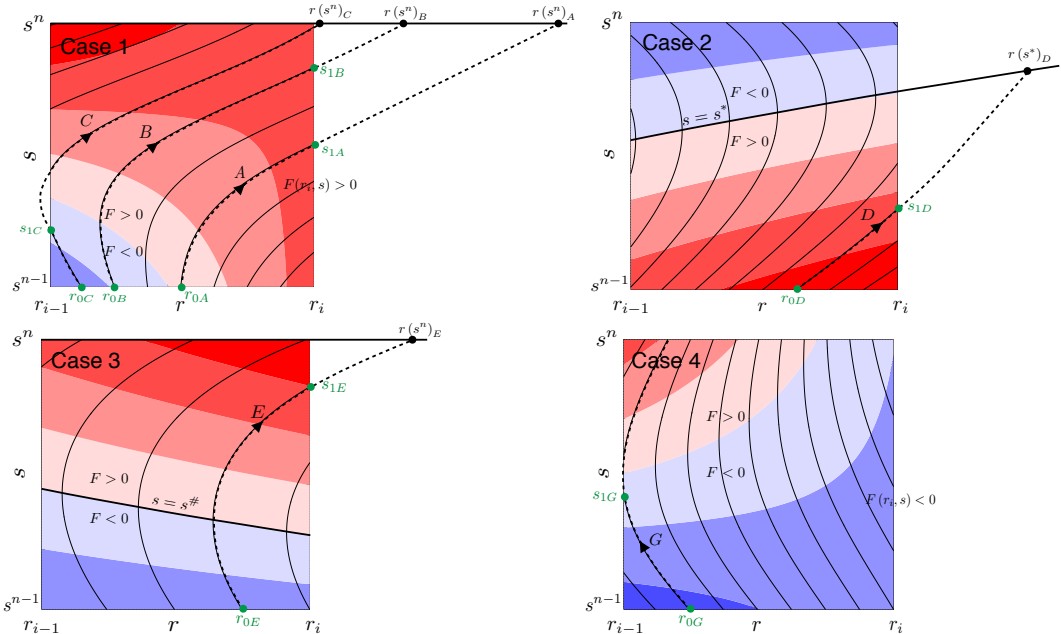

**Figure 3.** The four different cases of how trajectories might reach the grid face at $r = r_i$. Note that the trajectories for case 4 can not reach $r = r_i$. The background colours are the same as in Fig. 2 with $F > 0$ in red and $F < 0$ in blue. The dashed trajectories outside the grid box denote the necessary computed fictive paths for estimating when $s = s_1$ and if the trajectories reach $r_1(s_1) = r_i$.

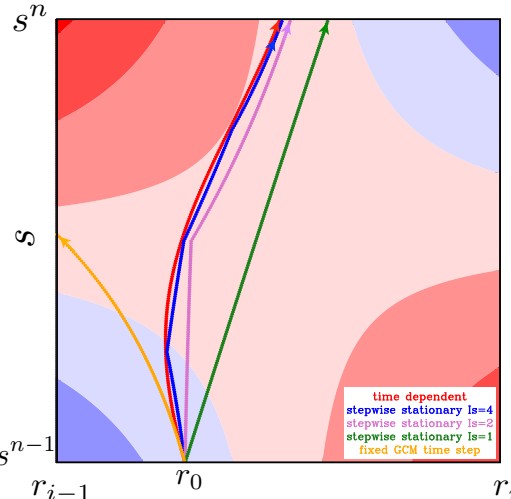

**Figure 4.** Example of how the trajectories differ when computed with the "fixed GCM time step" method in orange, the "stepwise-stationary" method in blue, purple and green as well as the "time-dependent" method in red. They all start at the same time $s^{n-1}$ and in the same position $r_0$ but exit the grid at different locations and times. Note that the "stepwise-stationary" method needs at least 4 intermediate time steps ($I_S$) to be close to "time-dependent" trajectory. The background colours are the same as in Fig. 2 with $F > 0$ in red and $F < 0$ in blue.

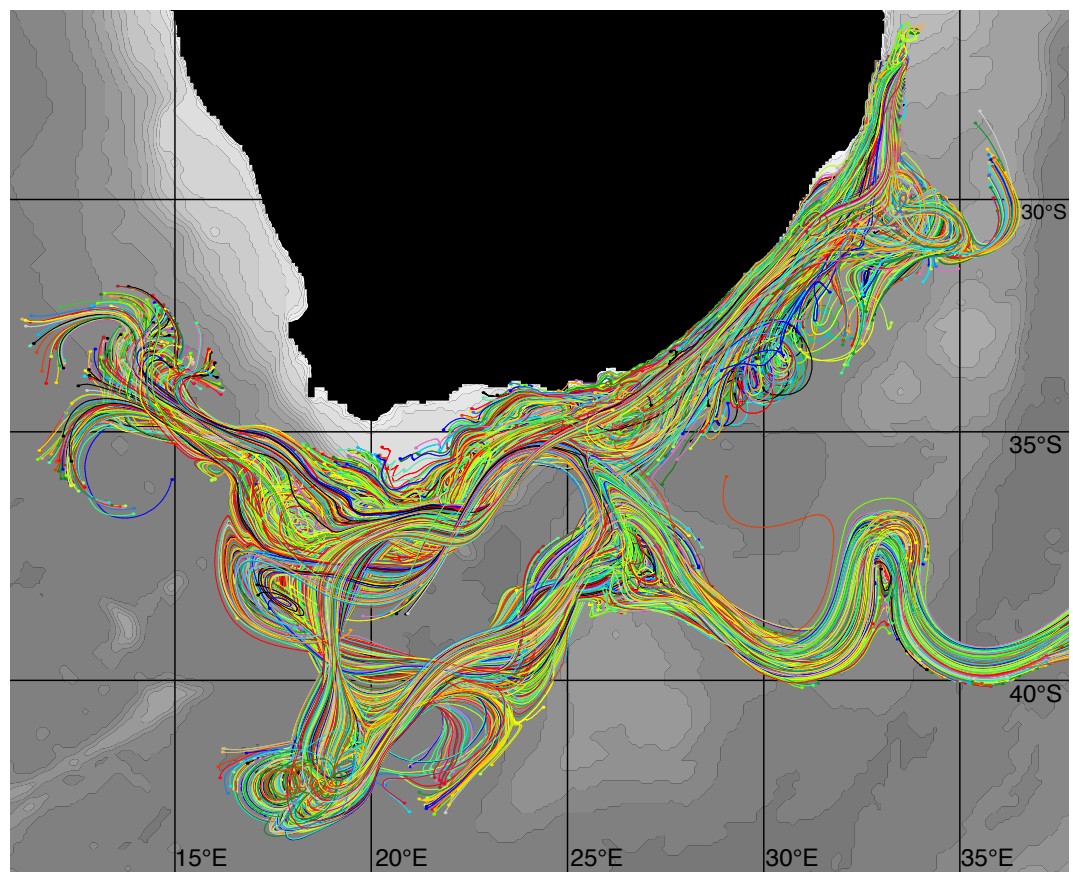

**Figure 5.** Agulhas trajectories started evenly distributed in a square of 4 grid cells and followed for 50 days. Colouring used to separate the trajectories from each other.

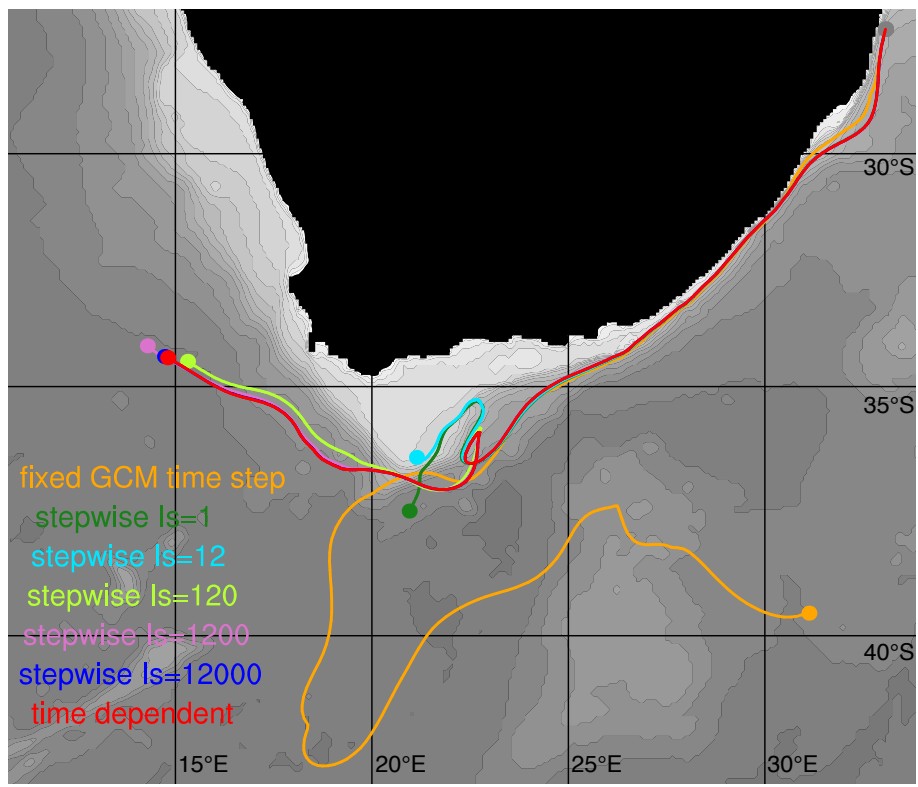

**Figure 6.** Example of ocean trajectory paths due to different trajectory schemes and number of intermediate time steps. The "time-dependent" method results in red and those obtained with the "stepwise-stationary" method with $I_S = 1, 12, 120, 1200$ and $12000$ as well as "fixed GCM time steps". Note that these homologous trajectories were selected to illustrate that "stepwise-stationary" trajectories are closer to "time-dependent" trajectories when the number of intermediate time steps ($I_S$) is increased.

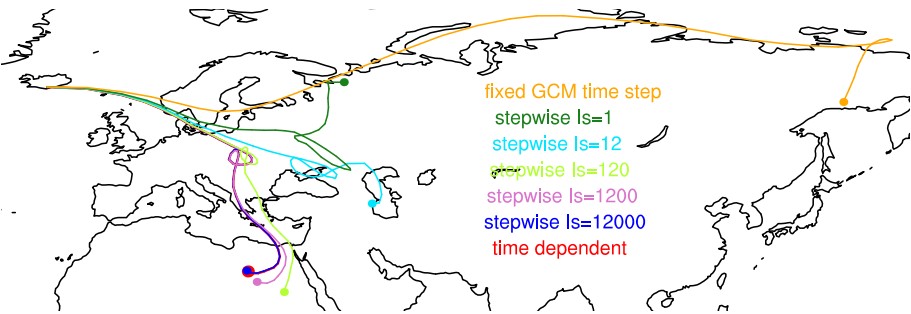

**Figure 7.** Example of atmospheric trajectory paths starting form the Eyjafjallajökull Volcano during it's eruption calculated with different trajectory schemes and number of intermediate time steps. Same colour coding of the trajectories as in Fig. 6. Note that the red "time-dependent" and the blue "stepwise" with $I_S = 12000$ trajectories are nearly identical.

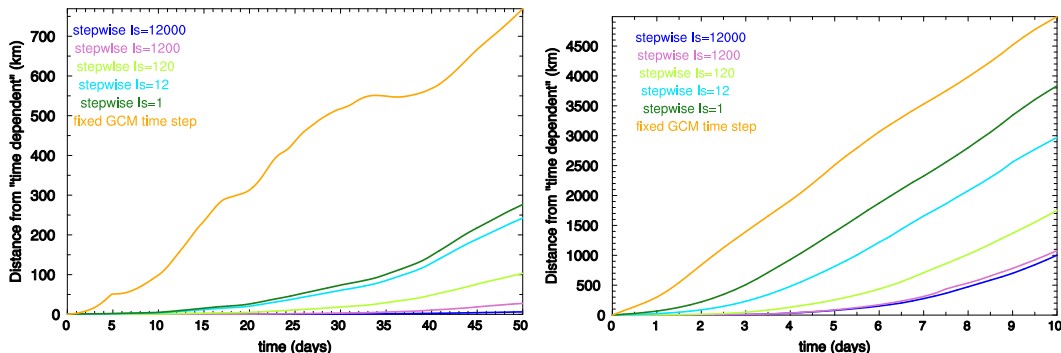

**Figure 8.** Average distance between the "time-dependent" trajectories and the "stepwise-stationary" ones for the different time-steps with $I_S = 1, 12, 120, 1200$ and $12000$ as well as "fixed GCM time steps". The left panel represents the ocean Agulhas trajectories and the right panel the atmospheric ERA-Interim ones. Note that the more intermediate steps used by the "stepwise-stationary" scheme the closer results to the "time-dependent" scheme.

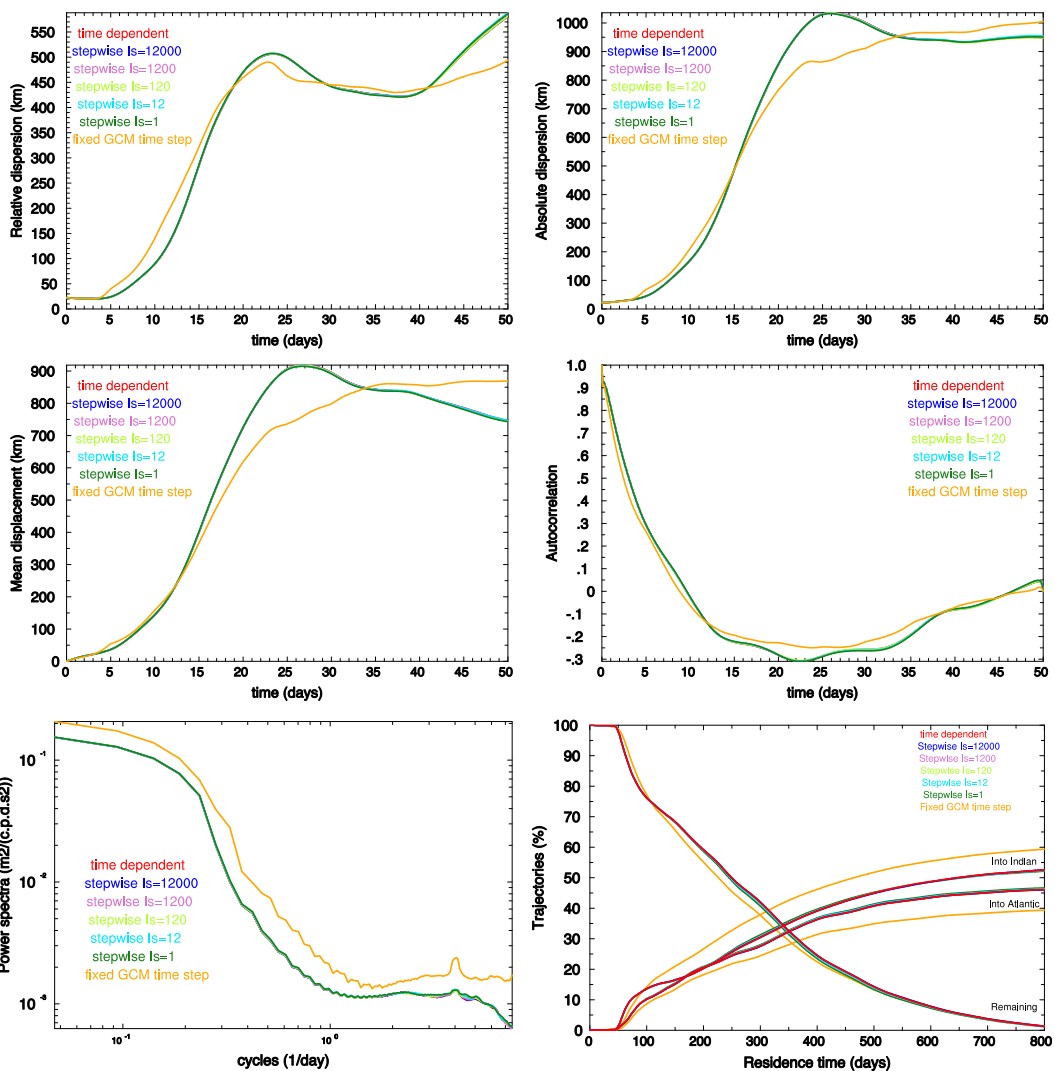

**Figure 9.** Lagrangian statistics of the ocean Agulhas trajectories. The relative dispersion (top left), the absolute dispersion (top right), the mean displacement travelled by the trajectory cluster (middle left), the average Lagrangian velocity autocorrelation of the trajectories (middle right). The average power spectra of the Lagrangian velocities (lower left). The residence time evolution of the trajectory particles in the Agulhas region. Note that all statistics show very similar results, where only those based on the "fixed GCM time step" (orange curves) differ from the rest.

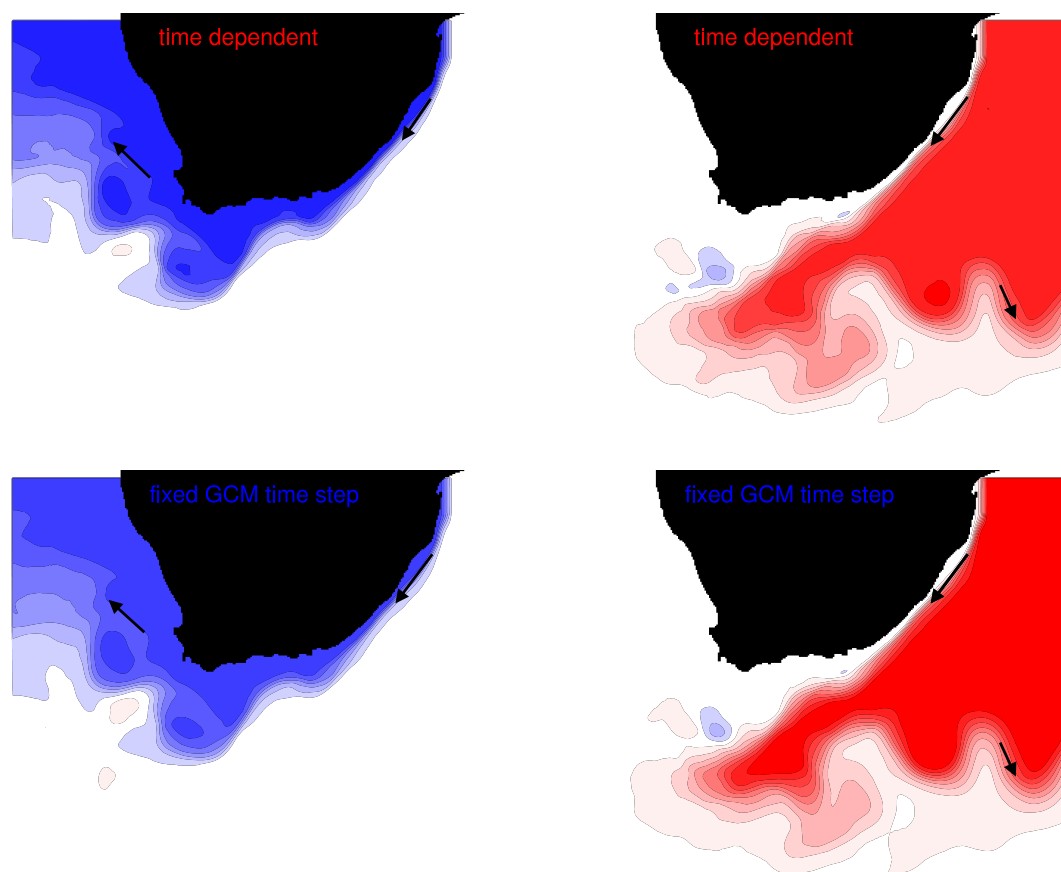

**Figure 10.** The Lagrangian decomposed barotropic stream function based on the particles released as previously but followed until they leave the Agulhas region into the Atlantic (left panels) or the Indan Ocean (right panels). The top panels with the "time-dependent" scheme and the lower panels with the "fixed GCM time step" scheme. Note that there is more water (one stream line extra) flowing into the Atlantic with the "time-dependent" scheme than with the "fixed GCM time steps" scheme, which instead favours relatively the flow into the Indian Ocean. Stream line intervals of 8 $Sv$ ($10^6\ m^3/s$).