# Peer review of "Evaluation of oceanic and atmospheric trajectory schemes in the TRACMASS trajectory model v6.0"

_Geoscientific Model Development, 2016_

## Referee Comment (RC1) · Anonymous Referee #1 · 9 Oct 2016

In this manuscript, the authors present a summary of the latest version of TRACMASS code, to be used to calculate trajectories of virtual particles in the ocean and atmosphere. In particular, they focus on the different time stepping schemes, explicitly comparing the skill and efficiency of stepwise stationary against linear interpolation schemes, to show that the latter is more accurate and faster (for the same accuracy) than the former.

In general, this is a well-written and interesting manuscript, that delves into a major question in the field: how to best calculate virtual particle trajectories. However, I have a number of major comments that I would like to see the authors address before I think the study could be published

1) Given that the authors have named this version v6.0, it would be good to summarise how it is different from the previous versions. What's new? In particular, almost all of the equations and derivation in section 2 were also presented in De Vries and Döös (2001). Given that this time-dependent option has apparently been in TRACMASS for so long, why focus on it now in version 6.0? If only the explicit comparison against the stepwise stationary algorithm for real-life applications such as NEMO and ERA-Interim is new, then this should be much clearer reflected throughout the manuscript (including the title)

2) The authors state on numerous occasions that their code is 'mass-conserving', but without actually proving this. Also, there is a key difference between 'mass conserving' and 'accurate'. A hypothetical scheme where particles are motionless and don't move will be mass-conserving, but not very accurate. The authors seem to conflate the two concepts a bit. Chu and Fan (2014) showed that this linear uncoupled interpolation scheme is in fact not the most accurate, so it would be good to see a response to that paper here. How does linear temporal interpolation affect mass conservation and accuracy, as well as the agreement with tracer fields (see also page 2, line 3)?

Minor comments:

- page 1, line 24: perhaps good to define what's meant with a 'grid cell' here. A model grid cell?

- page 2, line 14: What type of 'continuous interpolation' is meant? Spline? Linear?

- page 3, line 23: This discussion of mass and volume interchangeability in OGCMs of course is only true in hydrostatic models (also page 4, line 17).

- page 6, line 20: This comment about how TRACMASS works on any vertical grid has been made already, and there is probably no need to mention it again here

- page 8, line 6: Are there any physical interpretations for $\alpha$, $\beta$, $\gamma$ and $\delta$? In particular for $\alpha$, what kind of flow is $\alpha$

[Figure]

> 0 versus $\alpha$ < 0?

- page 10, line 14: The terminology of grid cell boundaries is a bit confusing at times. Here it is called a 'wall', even though it is not a land-sea boundary. I suggest to carefully go through the manuscript to standardise the wording used to distinguish ocean-ocean (or atmosphere-atmosphere) grid faces from land-ocean faces.

- page 10, line 27: Would be good to explicitly mention which root-solving algorithm is used.

- page 11, line 28: 'Conveyor Belt' is a simplistic term here, better to call it thermohaline circulation?

- page 11, line 29: How are the particles seeded in the vertical? At all depth levels?

- page 12: There appears to be no reference to Figure 5 in the text between the first references to Figure 4 and Figure 6?

- page 16: refer to Lacasce (2008) here, for the standard work on the statistics of particle dispersion in the ocean?

- Figure 4: What does the colouring of the trajectories represent? And it might also be useful to add a grid with selected longitudes and latitudes, so that reader unfamiliar with the Agulhas region can orient themselves (that latter point also for Figure 9)

- Figure 7: Beyond showing the mean distance, would it also be useful to show the spread (e.g. the one standard deviation of each line with time)?

- Figure 8: The presentation of Fig 8 is not ideal, because most lines fall on top of each other. I appreciate that this is the whole point of the Figure, but a quick reader might be confused where the other lines are. Is there no better way to show that 6 of the 7 lines essentially lay on top of each other?

Type-os etc:

- page 2, line 18: Replace 'always' with 'typically'?

- page 2, line 23: 'behind these can be found'

- page 5, line 23: should be Eq (19)?

- page 9, line 16: Eq 32 should not be part of this list?

- page 9, line 18: There is no following subsection, there is just the text below

- page 10, line 10: use 'domain' rather than '"box"'?

- page 10, line 28: 'r' at end of line misses subscript i

- page 11, line 20: 'implies that they'

- page 12, line 16: 'distances have been possible to compute since all'

———————————————

---

## Referee Comment (RC2) · S. M. Griffies (Referee) · 13 Oct 2016

This is a useful manuscript that documents some important numerical details related to Lagrangian trajectory analysis. The authors are leads for the TRACMASS code, and they bring decades of experience and leadership to the problem. I recommend publication. However, I ask that many details missing from the manuscript be exposed and discussed. Here are my specific requests along with minor edit points.

Note that I will focus my comments on those that complement those from the other reviewer. In particular, this reviewer raises an important point about distinction between "accurate" and mass conserving. It would be useful for the authors to discuss this point.

**GENERAL COMMENTS**

–As detailed here, and in the earlier literature, the TRACMASS approach performs an analytic integration of the trajectory within a grid cell. This point is emphasized in the present manuscript. Importantly, this integration is enabled by an \*\*assumption\*\* that the subgrid scale velocity components are linear functions of their corresponding directions: [u(x), v(y), w(z)]. Surprisingly, this critical assumption is not explicity noted in the present manuscript. It should in fact be emphasized and defended.

How/where will it break down? As written, words such as "the trajectory solutions are exact" (pg 5, line 10) make it look like TRACMASS is performing magic. Instead, it is following an exact treatment based on the assumption of subgrid [u(x), v(y), w(z)].

–The differential equations for the position within a grid cell are given by equation (17) for the stationary case, and equation (26) for the time-dependent case. Both equations are offered to the reader as if they should be an obvious consequence of something a priori. However, both equations need more build up to motivate and rationalize.

The only statement to suggest where equation (17) comes from is line 19 on pg 5:

"The transport and position within the grid box are now related by U = dr/dx...".

However, this is a statement that offers no motivation nor a derivation. What is the basis for this relation?

So as written, equations (17) and (26) seemingly appear from no where, and the reader is left scratching his/her head. Sans shared intuition, these equations remain mere black boxes to the reader, which is of no use to the reader.

–At the end of Section 2, I found myself wanting to see a clear schematic to summarize the stationary method and the time-dependent method. Likely these schematics appear in the basic literature. But given that you are rederiving the methods here, it would serve the reader well to have such schematics presented again, perhaps in an updated manner. These schematics could offer far more conceptual understanding

than the maths presented in Section 2.

–The word "this" is used many placed without qualifying. The reader is often left wondering what "this" referes to. Please be more careful with letting the reader know what "this" refers to. It is important to do so in order not to lose the reader, especially the novice.

SPECIFIC COMMENTS

pg 2, line 25: "This in contrast..." suggest changing to "This method contrasts to the..."

pg 3, line 11: change "an GCM" to "a GCM"

pg 4, equation (4): The grid cell thickness is a function of (i,j,k,n). However, this space-time dependence is not consistently displayed in the manuscript, such as in equation (8) where we only see dependence on (k,n). Where it is relevant, and where (i,j) are exhibited for Delta x and Delta y, please also display such dependence for Delta z.

pg 4, line 17: not all ocean GCMs are incompressible (e.g., MOM and MITgcm have non-Boussinesq compressible options). So please qualify this statement; e.g., "many ocean models are incompressible".

pg 4, equations (9), (10), (12): As written, these equations are not usable when integrating within the model code online, unless we have the n+1 value of the thickness, density, and/or pressure. So please comment on what operationally you mean by these equations.

pg 5, line 16: It is useful to here state that r = x/Delta x is defined separately for each direction and for each grid cell. Namely, there is no assumption that Delta x is uniform across the model.

pg 5, line 16: change "velocity" to "transport"

pg 6, lines 9-10: this sentence appears to be an orphan from another paragraph. Perhaps it should go into the previous paragraph, though with some editing to reduce

Interactive
comment

redundancy.

pg 6, line 30: remove "with" at start of line.

pg 6, line 30: Global GCMs are often run with time steps of **hours** to minutes. For example, the GFDL-CM2.1 ocean component, which is one degree, uses a 2hours time step. GFDL-CM2.5, which is 1/4 degree, uses 30minutes.

pg 7, line 11: I believe that "spatial resolution" should be changed to "spatial grid spacing". Namely, resolution is a pure number whereas grid spacing has units of length. You are stating (Delta x,Delta y,Delta z), which is grid spacing, not resolution.

pg 7, equation (25): which time level do you use for Delta z in the case where the grid cell thickness is a function of time? n? n-1? average?

pg 9, line 15: suggest "...is that in the time-dependent scheme, the transit times..."

pg 11, line 8 and title to Section 3.1: No one has proven that 1/12th degree **resolves** mesoscale eddies globally. Instead, such models **admit** mesoscale eddies, but they are surely not fully resolved globally. So to state the ORCA12 is "eddy resolving" is an overstatement that is not justified.

pg 12, line 32: what is "this" referring to in the middle of the line?

pg 13, line 17: "the the"

pg 13, line 32: "...quantify this **difference** we compared..."

pg 14, line 26: It is stated here that the time-dependent scheme is more accurate than the alternatives. I suspect that it is indeed more accurate. But I see no where in the paper where a "truth" is used to base this conclusion. Or did I miss something??

pg 15, lines 14, 16, 25: please specify what "this" refers to on each of these lines. The reader should not be asked to assume he/she knows what you are referring to.

Figure 1 caption. Remove first "blue" in the second sentence.

Figure 2 caption "the the"

END OF REVIEW

---

## Referee Comment (RC3) · Anonymous Referee #3 · 1 Nov 2016

This manuscript offers a summary of the schemes available in the TRACMASS Lagrangian analysis code and provides a comparison of their application to two numerical models. As numerical models increase in resolution it is important that the development of Lagrangian tools keeps pace, such that they can provide accuracy and computational efficiency. These are difficult to achieve in 'offline' Lagrangian analyses. The latest developments, and their usefulness as outlined here, are important and will be of use to the community.

The manuscript is well written, and I find the analyses to be useful and the figures to be helpful. While the derivations have already been provided in earlier work (such as the cited de Vries and Döös 2001 article), which risks making the paper unnecessarily

bulky, it is useful to be able to consider and compare the two different schemes in a single article and I do not object to it. Following some minor corrections and clarifications that I have listed below, I recommend the article for publication.

General considerations:

1) In section 3 it appears to be taken at face value that the time-dependent case represents a 'model truth', to which the other cases are compared. The sentence on P2 L15 suggests that it is "logical" to use a stepwise-stationary scheme that analyses output at the model time-integration frequency. Is there some way of clarifying/demonstrating that the time-dependent solution is the most realistic solution to the model transports? I suppose that a real 'truth' could be calculated by outputting the model at every model integration step and then performing the stepwise-stationary analysis. Since this is obviously very laborious, perhaps the authors can provide an easier description.

2) - P13: L26-29. In order to be able to better interpret those studies that have used the fixed timestep scheme, it would be useful to know if the values provided on e.g. P13 L26-29 are sensitive to the number of particles used and the time period over which the particles are seeded. Is it possible that the fixed GCM time step scheme converges (closer) towards the other schemes when a long(er) seeding period is used along with a large(r) number of particles? In this case, the streamfunctions could be compared to an Eulerian streamfunction that is taken as the actual model truth.

Similarly, do the number of particles used in the experiments constitute a "large ensemble".

3) In the absence of having tested a suite of models, I think the authors could be clearer in places (e.g. P15 L20) that their results are specific to the resolutions of their chosen models. While the increased accuracy of the method will certainly translate to a consideration of lower resolution models, the relative importance is likely going to decrease. Perhaps for certain applications the fixed timestep solution could be just as meaningful as the time-dependent one, if a large enough number of particles are

integrated?

Specific comments:

- P1 L3-4. I find a "limited period of time" to be a pretty vague description. Something more like "... stationary for set intervals of time between saved model outputs" might be clearer.

- P1 L13 At this point in the article, "more accurate" seems ambiguous as to whether it is more accurate w.r.t the time-dependent case or w.r.t itself. Perhaps a change to "increasingly more accurate" would make it clearer.

- P3 L12: superscript n is not defined until P4. Also, n is sometimes used as a subscript, presumably by mistake (e.g. P3 L24).

- P3 L30: It is not made clear why it is more advantageous or why the direct calculation would be any more accurate than is done by the model. Is it because of the interpolation that is applied by TRACMASS?

- P4 L7: If Tracmass can work on models in which a variable vertical resolution is also spatially dependent, as stated on P3, then should delta_z in equation 8 also have subscripts i,j?

- P4 L10: delta_t_G is defined only later on.

- P4 L14: I'd have thought convention normally has k=0 as the surface grid cell, not the bottom.

- P4 L17: It appears that a numbered list is started here, but I'm not sure why.

- equation 13: This has already been written of line 24 of the previous page. Accordingly, could equations 14 and 15 also be moved up to where that previous definition of hydrostatic balance is given, which seems a more natural place for these equations to go? Currently they are returning the discussion to horizontal velocities after having discussed vertical velocities.

- P5 L5-9: I feel this section would be more helpful if given at an earlier point in the paper, plus it is largely repeating what was stated on P3 L13.

- P5 L16: Perhaps say here that this is done for V and W too.

- P5 L28: Typo Eq (1).

- P6 L2. Regarding "if this is not the case", it is not clear whether this is referring to U(r1) or U(r2) being positive, or both.

- P7 L11-15: It should perhaps say here that both the fixed time step and the stepwise stationary cases will be tested.

- Equation 24: typo on the second line of the equation, in the first F term - a subscript n. Similar typo in equation 38.

- Figure 2: What is a 'region' here, and why only three of them in subplot (a)? Also, given that the stepwise-stationary method can also include temporal interpolation, it should be stated here that these solutions are for the time-dependent case.

- P9 L15: It is not clear whether "this case" is referring to the time-dependent or stepwise case.

- P9 L18. The "following subsection" or 'this subsection'?

- Section 3.1: There is no mention of Figure 5.

- Figures 5 onwards: Isn't $I\_s=1$ the same as the fixed GCM time step?

- P13 L2: It should perhaps be clarified that this is now referring to improvements in the GCM, not the Lagrangian model.

- P13 L18: Figure 9 shows neither a subtraction nor the stepwise-stationary case.

- P13 L20-21: I don't understand this sentence, which appears to contrast with those on lines 26-29 in the same paragraph.

- P14 L11: Instead of saying "for some time", which is ambiguous, I would suggest something more like "for the duration of a user defined intermediate time step between model output fields". Also, the use of the past tense here doesn't work well, especially since the next sentence is in the present again.

- P14 L13: Similarly, instead of "is in steady state" I would say something more like "is steady during each time step".

---

## Author Response (AR1)

**Response to referee 1.**

**Answers to the major comments**

1. The novelties of version 6.0 and the differences between this paper and Vries and Döös (2001) is now stated in the two first paragraphs in the introduction.

2. The reason for now citing and discussing the differences with Chu and Fan (2014) is that we simply do not agree with their approach and their results. This would require a separate study, which would be beyond the scope of this paper. In their experiment they fail to keep the trajectories along the stream lines for the Stommel Gyre, which TRACMASS is able to do with all its schemes. We have discussed together with Bruno Blanke to submit a note on this issue with the Chu and Fan (2014) paper.

**MINOR COMMENTS**

- page 1, line 24: perhaps good to define what's meant with a "grid cell" here. A model grid cell?

Answer: We have added the word "model"

- page 2, line 14: What type of "continuous interpolation" is meant? Spline? Linear?

Answer: We have added the word "linear".

- page 3, line 23: This discussion of mass and volume interchangeability in OGCMs of course is only true in hydrostatic models (also page 4, line 17).

Answer: We have rewritten this now making it clear it is only valid for models that are incompressible.

- page 6, line 20: This comment about how TRACMASS works on any vertical grid has been made already, and there is probably no need to mention it again here

Answer: We have removed this paragraph.

- page 8, line 6: Are there any physical interpretations for  $\alpha$ ,  $\beta$ ,  $\gamma$  and  $\delta$ ? In particular for  $\alpha$ , what kind of flow is  $\alpha > 0$  versus  $\alpha < 0$

Answer: We have added a physical interpretations for  $\alpha$  at the end of section 2.4.1 and 2.4.2

- page 10, line 14: The terminology of grid cell boundaries is a bit confusing at times. Here it is called a "wall", even though it is not a land-sea boundary. I suggest to carefully go through the manuscript to standardise the wording used to distinguish ocean-ocean (or atmosphere-atmosphere) grid faces from land-ocean faces.

Answer: We have replace the words "wall" and "grid-box wall" by "grid face" in the entire text.

- page 10, line 27: Would be good to explicitly mention which root-solving algorithm is used.

Answer: Done.

- page 11, line 28: "Conveyor Belt" is a simplistic term here, better to call it thermohaline circulation?

Answer: We do not agree with this. Any circulation in T-S space can be defined as thermohaline but the Agulhas rings flowing north into the Atlantic are part of a global circulation often referred to as the "Conveyor Belt".

- page 11, line 29: How are the particles seeded in the vertical? At all depth levels?

Answer: Yes, at all depths, which we have added in the text now.

- page 12: There appears to be no reference to Figure 5 in the text between the first references to Figure 4 and Figure 6?

Answer: Fig. 5 is now references to between Fig. 4 and Fig. 5.

- page 16: refer to Lacasce (2008) here, for the standard work on the statistics of particle dispersion in the ocean?

Answer: We are now citing Lacasce (2008) in this appendix.

- Figure 4: What does the colouring of the trajectories represent? And it might also be useful to add a grid with selected longitudes and latitudes, so that reader unfamiliar with the Agulhas region can orient themselves (that latter point also for Figure 9)

 $\mathbf{2}$

Answer: We have added "Colouring used to separate the trajectories from each other.

- Figure 7: Beyond showing the mean distance, would it also be useful to show the spread (e.g. the one standard deviation of each line with time)?

Answer: That is basically what the relative dispersion shows

- Figure 8: The presentation of Fig 8 is not ideal, because most lines fall on top of each other. I appreciate that this is the whole point of the Figure, but a quick reader might be confused where the other lines are. Is there no better way to show that 6 of the 7 lines essentially lay on top of each other?

Answer: We have tried different solution but the fact that the line lay on top of each other just reflects that they give very similar results.

Type-os etc: - page 2, line 18: Replace "always" with "typically"?

Answer: Done.

- page 2, line 23: "behind these can be found"

Answer: Changed.

- page 5, line 23: should be Eq (19)?

Answer: No.

- page 9, line 16: Eq 32 should not be part of this list?

Answer: Yes and it is written Eqs. (31)-(34), which includes Eq. 32.

- page 9, line 18: There is no following subsection, there is just the text below - page 10, line 10: use "domain" rather than "?box?"?

Answer: True and we have rewritten the text and deleted "following subsection".

- page 10, line 28: "r" at end of line misses subscript i

Answer: Added an index i to this r.

- page 11, line 20: "implies that they"

Answer: Changed as suggested

- page 12, line 16: "distances have been possible to compute since all"

Answer: The extra "been" has been removed.

**Response to referee Griffies**

**GENERAL COMMENTS**

1. As detailed here, and in the earlier literature, the TRACMASS approach performs an analytic integration of the trajectory within a grid cell. This point is emphasized in the present manuscript. Importantly, this integration is enabled by an \*\*assumption\*\* that the subgrid scale velocity components are linear functions of their corresponding directions: [u(x), v(y), w(z)]. Surprisingly, this critical assumption is not explicitly noted in the present manuscript. It should in fact be emphasized and defended.

How/where will it break down? As written, words such as "the trajectory solutions are exact" (pg 5, line 10) make it look like TRACMASS is performing magic. Instead, it is following an exact treatment based on the assumption of subgrid [u(x), v(y), w(z)].

Answer: We have added a few sentences on this in the first paragraph of section 2 and section 2.2 to better highlight this key assumption. We agree that the use of the word "exact" may mislead readers, and have rewritten a few sentences such as p5, line10 to emphasise that the trajectories are solutions to a differential equation, and that there is nothing magical about it.

2. The differential equations for the position within a grid cell are given by equation (17) for the stationary case, and equation (26) for the time-dependent case. Both equations are offered to the reader as if they should be an obvious consequence of something a priori. However, both equations need more build up to motivate and rationalize. The only statement to suggest where equation (17) comes from is line 19 on pg 5: "The transport and position within the grid box are now related by U = dr/dx...". However, this is a statement that offers no motivation nor a derivation. What is the basis for this relation?

So as written, equations (17) and (26) seemingly appear from no where, and the reader is left scratching his/her head. Sans shared intuition, these equations remain mere black boxes to the reader, which is of no use to the reader.

Answer: We have rewritten the first paragraphs of section 2.2 (stationary scheme) and 2.4 (time-dependent scheme) to better lead up to the differential equations that we use to calculate trajectories. We hope this is clearer to the reader.

3. At the end of Section 2, I found myself wanting to see a clear schematic to summarize the stationary method and the time-dependent method. Likely these schematics appear in the basic literature. But given that you are rederiving the methods here, it would serve the reader well to have such schematics presented again, perhaps in an updated manner. These schematics could offer far more conceptual understanding than the maths presented in Section 2.

Answer: A very good point and we have both added a paragraph at the end of section 2 and a figure, summarising the resulting differences between the schemes within a time-space cell.

4. The word "this" is used many placed without qualifying. The reader is often left wondering what "this" referes to. Please be more careful with letting the reader know what "this" refers to. It is important to do so in order not to lose the reader, especially the novice.

Answer: We have rewritten a number of sentences in order to remove "this".

**MINOR COMMENTS**

- page 1, line 24: perhaps good to define what's meant with a "grid cell" here. A model grid cell?

Answer: Yes and we have now written "model grid cell".

 $\mathbf{6}$

**RESPONSE TO REFEREE 3.**

**GENERAL CONSIDERATIONS**

1. In section 3 it appears to be taken at face value that the time-dependent case represents a "model truth", to which the other cases are compared. The sentence on P2 L15 suggests that it is "logical" to use a stepwise-stationary scheme that analyses output at the model time-integration frequency. Is there some way of clarifying/demonstrating that the time-dependent solution is the most realistic solution to the model transports? I suppose that a real "truth" could be calculated by outputting the model at every model integration step and then performing the stepwise-stationary analysis. Since this is obviously very laborious, perhaps the authors can provide an easier description.

Answer: We have now included the following sentences in the discussion:"We thus conclude that the "time-dependent" scheme is the most accurate of those tested here for two reasons. Firstly for theoretical reasons since the "time-dependent" scheme does not assume stationary velocities during any period of time. Secondly the trajectories computed with the "stepwise-stationary" scheme converge towards those computed with the "time-dependent" scheme for increasing numer of intermediate time steps. A future study could be to calculate trajectories first using fields stored at each GCM time step and second using fields stored at longer time intervals. In the first case, trajectories would be very accurate and could represent a "truth", and the second case could be used to evaluate which of the two schemes is the closest to the "truth". "

2. P13: L26-29. In order to be able to better interpret those studies that have used the fixed timestep scheme, it would be useful to know if the values provided on e.g. P13 L26-29 are sensitive to the number of particles used and the time period over which the particles are seeded. Is it possible that the fixed GCM time step scheme converges (closer) towards the other schemes when a long(er) seeding period is used along with a large(r) number of particles? In this case, the streamfunctions could be compared to an Eulerian streamfunction that is taken as the actual model truth. Similarly, do the number of particles used in the experiments constitute a "large en- semble".

Answer: We have added at the end of section 3.4: "We have repeated the above oceantrajectory experiment by releasing the particles in other time periods and increasing the ensemble size. The results only changed marginally."

3) In the absence of having tested a suite of models, I think the authors could be clearer in places (e.g. P15 L20) that their results are specific to the resolutions of their chosen models. While the increased accuracy of the method will certainly translate to a consideration of lower resolution models, the relative importance is likely going to decrease. Perhaps for certain applications the fixed timestep solution could be just as meaningful as the time-dependent one, if a large enough number of particles are integrated?

Answer: "We have here only tested one OGCM and one AGCM simulation, but we speculate that at coarser resolution in both space and time, the differences obtained with the two schemes would increase. However, in non eddying simulations (e.g. 1° ocean models) this may not be true due to the low variability of the flow."

**Specific comments**

- P1 L3-4. I find a "limited period of time" to be a pretty vague description. Something more like "... stationary for set intervals of time between saved model outputs" might be clearer.

Answer: Done

- P1 L13 At this point in the article, "more accurate" seems ambiguous as to whether it is more accurate w.r.t the time-dependent case or w.r.t itself. Perhaps a change to "increasingly more accurate" would make it clearer.

Answer: Done

- P3 L12: superscript n is not defined until P4. Also, n is sometimes used as a subscript, presumably by mistake (e.g. P3 L24).

Answer: n is now defined where it first used.

- P3 L30: It is not made clear why it is more advantageous or why the direct calculation would be any more accurate than is done by the model. Is it because of the interpolation that is applied by TRACMASS?

Answer: It is because the TRACMASS trajectory schemes rely on mass continuity. Ideally, the two methods should give the same result.

- P4 L7: If Tracmass can work on models in which a variable vertical resolution is also spatially dependent, as stated on P3, then should  $\Delta_z$  in equation 8 also have subscripts i, j?

Answer: Yes it should. We have added this in all equations where  $\Delta z$  is written.

- P4 L10:  $\Delta t_G$  is defined only later on.

Answer: We have now defined it.

- P4 L14: I'd have thought convention normally has k=0 as the surface grid cell, not

the bottom.

Answer: Yes, but in TRACMASS we redefine the index k in order to have increasing index with positive vertical velocity, which makes the code simpler.

- P4 L17: It appears that a numbered list is started here, but I?m not sure why.

Answer:Removed

- equation 13: This has already been written of line 24 of the previous page. Accordingly, could equations 14 and 15 also be moved up to where that previous definition of hydrostatic balance is given, which seems a more natural place for these equations to go? Currently they are returning the discussion to horizontal velocities after having discussed vertical velocities.

Answer: A good point. We have moved equations as suggested.

- P5 L5-9: I feel this section would be more helpful if given at an earlier point in the paper, plus it is largely repeating what was stated on P3 L13.

Answer: We have removed this paragraph and added some text in the beginning of the section instead.

- P5 L16: Perhaps say here that this is done for V and W too.

- P5 L28: Typo Eq (1).

Answer: Done.

Answer: We have rewritten this in order to introduce the meridional and vertical displacements.

- P6 L2. Regarding "if this is not the case", it is not clear whether this is referring to U(r1) or U(r2) being positive, or both.

Answer: We have rewritten this.

- P7 L11-15: It should perhaps say here that both the fixed time step and the stepwise stationary cases will be tested.

Answer: We have added at the end of this section: "These two schemes together with a truly time dependent scheme, described in next section, will be tested."

- Equation 24: typo on the second line of the equation, in the first F term - a subscript n.

Similar typo in equation 38.

Answer: Corrected in both equations.

- Figure 2: What is a "region" here, and why only three of them in subplot (a)? Also, given that the stepwise-stationary method can also include temporal interpolation, it should be stated here that these solutions are for the time-dependent case.

Answer: We now state that this is for the "time-dependent" scheme and we use the word "corner" instead of "region".

- P9 L15: It is not clear whether "this case" is referring to the time-dependent or stepwise case.

Answer: We have rephrased this in order to clarify differences between the two cases.

- P9 L18. The "following subsection" or "this subsection"?

Answer: This sentence has been rephrased.

- Section 3.1: There is no mention of Figure 5.

Answer: We now mention this figure, which is now Fig. 6.

- Figures 5 onwards: Isn't  $I_s = 1$  the same as the fixed GCM time step?

Answer: No.  $I_S = 1$  is one average between two GCM outputs.

- P13 L2: It should perhaps be clarified that this is now referring to improvements in the GCM, not the Lagrangian model.

Answer: This has been clarified and the resolution regards the GCM and the sub-grid parameterisation the Lagrangian model, which was not clear.

- P13 L18: Figure 9 shows neither a subtraction nor the stepwise-stationary case.

Answer: We have this corrected this, which was due to that we had originally other stream functions.

- P13 L20-21: I don't understand this sentence, which appears to contrast with those on lines 26-29 in the same paragraph. C4

Answer: Same correction as for your previous comment.

- P14 L11: Instead of saying "for some time", which is ambiguous, I would suggest something more like "for the duration of a user defined intermediate time step between model output fields". Also, the use of the past tense here doesn't work well, especially since the next sentence is in the present again.

Answer: Thank you for this sentence, which we have now used.

- P14 L13: Similarly, instead of "is in steady state" I would say something more like "is steady during each time step".

Answer: We have changed to "The "time-dependent" scheme does not assume that the velocity is in steady state during any time interval since it solves the differential equations of the trajectory path not only in space but also in time."

**Evaluation of oceanic and atmospheric trajectory schemes in the TRACMASS trajectory model v6.0**

Kristofer Döös1, Bror Jönsson2, and Joakim Kjellsson3

1Department of Meteorology, Stockholm University, SE-10691 Stockholm, Sweden.

2Department of Geosciences, Princeton University, Guyot Hall, Princeton, NJ 08544, USA

3Atmospheric, Oceanic, and Planetary Physics, University of Oxford, UK

Correspondence to: Kristofer Döös (doos@misu.su.se)

Abstract. Two Three different trajectory schemes for oceanic and atmospheric general circulation models are compared in two different experiments. The theories of the two trajectory schemes are presented showing the differential equations they solve and why they are mass conserving. One scheme assumes that the velocity fields are stationary for a limited period of time set intervals of time between saved model outputs and solves the trajectory path from a differential equation only as a

- 5 function of space, i.e. "stepwise stationary". The second scheme uses a is a special case of the "stepwise-stationary" scheme, where velocities are assumed constant between GCM outputs, it uses hence a "fixed GCM time step". The third scheme uses a continuous linear interpolation of the fields in time and solves the trajectory path from a differential equation as a function of both space and time, i.e. "time-dependent". A special case of the "stepwise-stationary" scheme, when velocities are assumed constant between GCM outputs, is also considered, named "fixed GCM time step". 
[revised manuscript text omitted]
 g is the gravitational acceleration and the pressure difference  $\Delta p$  between the bottom and top of the grid box is obtained using the hydrostatic approximation:

$$\Delta p_{i,j,k}^n = \rho_{i,j,k}^n \, g \, \Delta z^n \, .$$

The mass transports through the lateral grid walls in the AGCM expressed by Eqs. (1, 2) will also use Eq. (5) to determine  $\Delta z$ 20 and hence become-

$$\frac{U_{i,j,k}^n = u_{i,j,k}^n \Delta y_{i,j} \Delta p_{i,j,k}^n/g}{V_{i,j,k}^n = v_{i,j,k}^n \Delta x_{i,j} \Delta p_{i,j,k}^n/g}.$$

TRACMASS can handle the following different sorts of vertical coordinates: 1) depth-level models, 2) sigma-coordinate models, where the thickness depends on the total depth, which varies in each horizontal grid point, 3) *z*-star coordinates,

25 where the layer thicknesses depend on sea surface elevation, 4) isopycnal models, where  $\Delta z$  is the density layer thickness, an approach that was first used in TRACMASS by ? and 5) pressure and hybrid vertical coordinates for AGCMs as introduced by ?Eq. 5 has been used.

**2.2 The stationary case**

This scheme assumes that the velocity and pressure fields are in steady state and was introduced by ? and used and developed for ocean mass transport studies by ?. The velocity inside a grid cell is found by assuming that it is only a function of its direction, i.e. u = u(x), v = v(y), w = w(z). Linear interpolation gives the zonal velocity.

5
$$u(x) = u_{i-1,j,k} + \frac{x - x_{i-1}}{\Delta x} (u_{i,j,k} - u_{i-1,j,k})_{\pm},$$
 (16)

We know that and similarly for v(y) and w(z). To calculate the zonal position, x, of a trajectory, we use u = dx/dt, and ean write this write Eq. 16 as the differential equation

$$\frac{dx}{dt} - \underline{\underline{x}} \frac{u_i - u_{i-1}}{\Delta x} \underline{\underline{x}} + \frac{x_{i-1}}{\Delta x} (u_i - u_{i-1}) - u_{i-1} = 0.$$

If we now substitute x for a non-dimensional position  $r = x/\Delta x$   $r \equiv x/\Delta x$  and t for a scaled time  $s \equiv t/(\Delta x_{i,j}\Delta y_{i,j}\Delta z_k)$ , we get

$$\frac{dr}{ds} + \beta r + \delta = 0, \tag{17}$$

where F = dr/ds is the zonal volume or mass flux, and  $\beta \equiv F_{i-1,j,k} - F_{i,j,k}$  and  $\delta \equiv -F_{i-1,j,k} - \beta r_{i-1}$  are constants. Its solution describes the zonal displacement within the grid box between the walls faces and is found using the initial condition  $r(s_0) = r_0$  of its zonal position so that

15
$$r(s) = \left(r_0 + \frac{\delta}{\beta}\right)e^{-\beta(s-s_0)} - \frac{\delta}{\beta}.$$
 (18)

The scaled time  $s_1$  becomes

10

$$s_1 = s_0 - \frac{1}{\beta} \log \left[ \frac{r_1 + \delta/\beta}{r_0 + \delta/\beta} \right],\tag{19}$$

where  $r_1 = r(s_1)$  is given by either  $r_{i-1}$  or  $r_i$ , when a trajectory enters the western or eastern wallgrid face, respectively. The logarithmic factor in Eq. (419) can be expressed as  $\log[F(r_1)/F(r_0)]$ .

- For a trajectory reaching the wall grid face  $r = r_i$ , for instance, the transport or  $r = r_{i-1}$  both  $F(r_1)$  must necessarily be positive, so and  $F(r_0)$  must be of the same sign in order for Eq. (19) to have a solution, the transport. If  $F(r_1)$  and  $F(r_0)$  must also be positive. If this is not the case, then the trajectory either reaches the opposite wall at  $r_{i-1}$  or the signs of the transports are such that are of opposite signs there is a zero zonal transport somewhere inside the grid box, which at a position between  $r_1$  and  $r_0$  and this position is reached exponentially slowlyslow.
- The calculations of above procedure is repeated for meridional and vertical displacements, where now  $r = y/\Delta y$  or  $r = z/\Delta z$ . This yields non-dimensional position,  $r_1$ , and scaled time,  $s_1$  are performed determining, for the zonal, meridional and vertical displacements of the trajectory, respectively, inside the grid box under consideration. The smallest transit time  $s_1 - s_0$  and the corresponding  $r_1$  denote through which wall grid face of the grid box the trajectory will exit and move into the adjacent one. The exact displacements in the other two directions are then computed using the smallest  $s_1$  in the corresponding Eq. (18).

The solutions in the meridional and vertical directions are calculated similarly as the zonal one but using the meridional and vertical transports, respectively.

Note that Eqs. (18)-(19) are not valid if the transport fields across the grid box are constant, i.e. when  $(F_{i-1,j,k} = F_{i,j,k})$ , since it would imply a division by zero with  $\beta = 0$  in both equations. The differential equation then simplifies to

$$\quad \frac{dr}{ds} + \delta = 0, \tag{20}$$

which has the solution

$$r(s) = -\delta(s - s_0) + r_0, \tag{21}$$

and the scaled time  $s_1$  is

$$s_1 = s_0 - \frac{r_1 - r_0}{\delta}.$$
 (22)

10 The solution above allows  $\Delta z$  to vary in space and time. Hence, TRACMASS works for any generalised vertical coordinate system, e. g. z, z\*,  $\tilde{z}$ , or  $\sigma$ . ? used TRACMASS with ERA-Interim reanalysis, which uses terrain-following hybrid coordinates. 
[revised manuscript text omitted]

$$\frac{\beta\gamma - \alpha\delta}{\alpha} = \frac{F_{i,n}F_{i-1}^{n-1} - F_i^{n-1}F_{i-1}^n}{F_i^n - F_{i-1}^n - F_i^{n-1} + F_{i-1}^{n-1}} \frac{F_i^n F_{i-1}^{n-1} - F_i^{n-1}F_{i-1}^n}{F_i^n - F_{i-1}^n - F_i^{n-1} + F_{i-1}^{n-1}},$$
(38)

$$\frac{\gamma}{\alpha} = \frac{F_{i-1}^n - F_{i-1}^{n-1}}{F_i^n - F_{i-1}^n - F_i^{n-1} + F_{i-1}^{n-1}} - r_{i-1},$$
(39)

$$\xi = \frac{F_{i-1}^{n-1} - F_i^{n-1} + \alpha(s - s^{n-1})}{\sqrt{2\alpha}}, \tag{40}$$

25

$$\zeta = \frac{F_{i-1}^{n-1} - F_i^{n-1} + \alpha(s - s^{n-1})}{\sqrt{-2\alpha}}.$$
(41)

As above, s is the scaled time. The coefficient in Eq. (38) appearing in Eqs. (31) and (33) is exactly zero when either the  $r_{i-1}$  or  $r_i$  wall-grid face represents a solid boundary, so that transport  $F_i$  or  $F_{i-1}$  is zero for all n, respectively. In these instances,

the opposite wall grid face fixes  $r_1$ , and the root  $s_1 > s_0$  can be computed analytically. If there is no solution, we take  $s_1 = s^n$ . When all three transit times equal  $s^n$ , the trajectory will not move into an adjacent grid box but will remain inside the original one. Its new position is subsequently determined, and the next time interval is considered.

- The roots of Eq. (37) have to be computed numerically if  $(\beta\gamma \alpha\delta)/\alpha \neq 0$ . This is also true for locating the extrema of 5 the solutions given by Eqs. (31) and (33). Alternatively, one can consider the case F(r,s) = 0 using Eq. (24) to analyse where possible extrema are located. It follows that in the *s*-*r*-plane, the extrema lie on a hyperbola of the form r = (as + b)/(c + ds). Obviously, only the parts defined by  $s^{n-1} \leq s \leq s^n$  and  $r_{i-1} \leq r \leq r_i$  are relevant. Depending on which parts of the hyperbola, if any, lie in this "box" and satisfy the initial condition  $r(s_0) = r_0$ , the trajectory r(s) exhibits none, one, or at most two extrema. In the latter case, the trajectory will not cross either the wall grid face at  $r_{i-1}$  or the one at  $r_i$  (see Fig. 2 for an example). Hence,
- 10 the trajectories r(s) determining the transit time  $s_1 s_0$  will have at most one extremum, i.e., there is at most one change of sign in the local transport *F*.

An efficient way of proceeding is as follows: first consider the wall grid face at  $r_i$ . For a trajectory to reach this wallgrid face, the local transport must be nonnegative, which depends on the signs of the transport  $F_{i-1,n}$  and  $F_{i,n}$ ,  $F_{i-1,n}^n$  and  $F_{i,n}$ . Four distinct configurations may arise between the model outputs ( $s^{n-1} < s < s^n$ ), where the calculation of the trajectory takes place:

15 1.
$$F(r_i, s) > 0$$
 for  $s^{n-1} < s < s^n$ .

- 2. The sign of  $F(r_i, s)$  changes from positive to negative at  $s = s^*$ , where  $s^{n-1} < s^* < s^n$
- 3. The sign of  $F(r_i, s)$  changes from negative to positive at  $s = s^{\#}$ , where  $s^{n-1} < s^{\#} < s^n$ .
- 4.  $F(r_i, s) < 0$  for  $s^{n-1} < s < s^n$ .

These four cases are illustrated by the four panels of Fig. 3.

- For case 1, we evaluate r(sn) using the appropriate analytical solution. If, in addition r(sn) ≥ ri, then the trajectory has crossed the grid-box wall-face r = ri at s1 ≤ sn as shown by the trajectories A, B and C in Fig. 3. If the initial transport F(r0, s0) < 0, the trajectory may have crossed the opposite wall-grid face at an earlier time as illustrated by trajectory C in Fig. 3. This is only possible if case 3 applies for the wall-grid face at ri-1 and s# > s0, in which case it is determined whether r(s#) ≤ ri-1. If this is not the case, there is a solution to r(s1) r1 = 0 for r1 = ri and s0 < s1 ≤ sn. Subsequently, this root can be calculated numerically using a root-solving algorithm (?). But if r(sn) < ri or, if applicable, r(s#) ≤ ri-1, we proceed

[revised manuscript text omitted]

We have here computed the barotropic Lagrangian stream function from the released particles. The top left panel of Fig. 9 shows this computed with the "time-dependent" trajectory scheme. In order to measure the differences due to the the The influence of the the different trajectory schemes , we have subtracted the stream function obtained from on the inter-ocean

- 20 exchange of water masses, which takes place in the Agulhas region, has been evaluated by calculating Lagrangian stream functions. Fig. 9 shows the Lagrangian barotropic stream function computed from trajectories using the "time-dependent" trajectories from those integrated with scheme and the "stepwise-stationary" and "fixed GCM time step" trajectories (Fig. 9). It is only the scheme. Lagrangian decomposition has been used to compute two separate stream functions for each scheme, one from trajectories entering the Atlantic and one from those returning back into the Indian Ocean via the Agulhas retroflection
- 25 region. The "time-dependent" scheme favours slightly (one additional stream line) the entering into the Atlantic compared to the "fixed GCM time step" stream function that clearly deviates. The total transport between the different basins will, however, not differ much between the schemes. scheme. This is also clearly visible when computing the residence time, i.e. the time a trajectory stays trajectories stay within the Agulhas region. We have also computed the total amount of trajectories remaining in the Agulhas region as a function of time as as shown in the lower righthand panel of Fig. 8. We have also decomposed

[revised manuscript text omitted]
{1}{N-1} \frac{1}{M-1} \sum_{\underline{n=1}} \sum_{\underline{n=1}} \sum_{\underline{n=1}} \sum_{i=1}^{N-1} \left( x_{\underline{i},\underline{i},\underline{m}}^n(t) - \hat{x}_{\underline{i},\underline{i},\underline{m}}^n(t) \right)^2.$$
(A1)

It is hence the distance between the two trajectories  $x_i^n(t)$  and  $\hat{x}_i^n(t)x_{j,m}(t)$  and  $\hat{x}_{i,m}(t)$ , where t is the time, N-M the total number of trajectories of the cluster and i the spatial coordinate index (*i.e.* the zonal, meridional or vertical position of the nm-th trajectory  $x_i^n(t)x_{j,m}(t)$ ). The two trajectories  $x_i^n(t)$  and  $\hat{x}_i^n(t) \cdot x_{j,m}(t)$  and  $\hat{x}_{j,m}(t)$  will have the same initial position  $(x_i^n(t_0) = \hat{x}_i^n(t_0)) \cdot (x_{i,m}(t_0) = \hat{x}_{i,m}(t_0))$  but will then evolve differently since different trajectory schemes are used to compute their paths. In the present study, we only consider the horizontal dispersion. The vertical dispersion is, however, an important measure of the vertical mixing in the ocean but beyond the scope of the present study.

The mean position of the trajectory cluster is defined as

The *relative dispersion* is defined as the mean-square displacement of the trajectories relative to the time-evolving mean position:

$$D_{R}^{2}(t) \equiv \frac{1}{N-1} \frac{1}{M-1} \sum_{\substack{n=1 \ m=1}}^{N} \sum_{i=1}^{M} \left( x_{\underline{i} \ \underline{i}, \underline{m}}^{n}(t) - \overline{x_{i}(t)} \right)^{2}.$$
(A3)

The *absolute dispersion* is defined in the same way, but relative to the initial position of the cluster:

5
$$D_A^2(t) \equiv \frac{1}{N-1} \frac{1}{M-1} \sum_{\substack{n=1 \ m=1}}^{N-1} \sum_{i=1}^{N-1} \left( x_{\underline{i} \ \underline{i}, \underline{m}}^n(t) - \overline{x_i(t_0)} \right)^2,$$
 (A4)

where  $t_0$  is the initial time of the trajectory.

The mean displacement is defined as the displacement from the origin as a function of time

$$D_D(t_{\underline{n}}) \equiv \frac{1}{\underline{N}} \frac{1}{\underline{M}} \sum_{m=1}^{N} \sqrt{\sum_{i=1}^{2} \left[ x_{i,n}(t_n) - x_{i,n}(t_0) \right]^{2M}} \sqrt{\sum_{i=1}^{2} \left[ x_{i,m}(t) - x_{i,m}(t_0) \right]^{2}}.$$
(A5)

The Lagrangian velocity is obtained by using a non-centered finite difference:

$$\quad u_{i,m}(t_{\underline{n}}^{n}) \equiv \frac{dx_{i,m}(t_{n})}{dt} \frac{dx_{i,m}(t^{n})}{\sqrt{dt}} \approx \frac{x_{i,m}(t_{n}) - x_{i,m}(t_{n-1})}{t_{n} - t_{n-1}} \frac{x_{i,m}(t^{n}) - x_{i,m}(t^{n-1})}{\frac{t^{n} - t^{n-1}}{\sqrt{dt}}},$$
(A6)

with the same indices as before wihere n is the time level. Similarly, the acceleration was calculated by finite differencing of the velocity:

$$a_{i,m}(t_{\underline{n}}) \equiv \underbrace{\frac{du_{i,m}(t_n)}{dt}}_{dt} \underbrace{\frac{du_{i,m}(t^n)}{dt}}_{dt} \approx \frac{u_{i,m}(t_n) - u_{i,m}(t_{n-1})}{t_n - t_{n-1}} \underbrace{\frac{u_{i,m}(t^n) - u_{i,m}(t^{n-1})}{t_n - t_{n-1}}}_{(A7)}.$$

Note how velocity is not defined at the first position, and acceleration is not defined at the first velocity.

15 The *Lagrangian velocity autocorrelation* describes the correlation of the velocity at one time with that of previous times. The definition is

$$R(\tau) = \frac{\sigma^2(\tau)}{\sigma^2(\tau=0)} \approx R_q = \frac{\sigma_q^2}{\sigma_0^2} (t^q) \frac{(\sigma(t^q))^2}{(\sigma(t^0))^2}$$
(A8)

where  $\sigma^2(\tau)$  and  $\sigma^2(\tau=0)$  are the Lagrangian velocity auto-covariances for time lag  $\tau$  and no lag, respectively. q is the discrete time step and  $R_q$  is the autocorrelation at time step q.  $\sigma^2(\tau)$  is defined as

$$\quad \sigma^{2}(\tau) = \lim_{T \to \infty} \frac{1}{T} \int_{0}^{T} \mathbf{u}'(t+\tau) \cdot \mathbf{u}'(t) \, dt \approx (\sigma_{\underline{q}}(\underline{t^{q}}))^{2} \equiv \sum_{i=1}^{2} \frac{1}{N-q-1} \sum_{n=1}^{N-q-1} u'_{\underline{i,ni}(\underline{t^{n}})} u'_{\underline{i,n+qi}(\underline{t^{n+q}})}, \tag{A9}$$

where  $u'_{i,n} = u_{i,n} - \overline{u}_i u'_i(t^n) = u_i(t^n) - \overline{u}_i$  and  $\overline{u}_i$  is a time average of the segment. Note that the total velocity autocovariance is the sum of the zonal and meridional components,  $\sigma^2 = \sigma_{i=1}^2 + \sigma_{i=2}^2$ .

The Lagrangian time scale is defined as

5

$$T_L = \int_0^\infty R(\tau) \, d\tau. \tag{A10}$$

This is a measure of the *memory* of a trajectory, i.e. the time lag during which the Lagrangian velocity is correlated. When computing this integral, the point where  $R(\tau) = 0$  for the first time is used here as upper bound. This truncation is perhaps the most commonly used, due to the often noisy character of the auto-correlation function,  $R(\tau)$  for large  $\tau$ .

Acknowledgements. The authors wish to thank Peter Lundberg for constructive comments. This work has been financially supported by the Bolin Centre for Climate Research and by the Swedish Research Council. Joakim Kjellsson is supported by the UK Natural Environment Research Council grant NE/K012150/1: "Poles apart: why has Antarctic sea ice increased, and why can't coupled climate models reproduce

10 observations?". The GCM integrations and the trajectory computations were performed using resources provided by the Swedish National Infrastructure for Computing (SNIC) at the National Supercomputer Centre at Linköping University (NSC).

**Table 1.** The table shows the average distance between the "time-dependent" integrated trajectories and the "stepwise-stationary" integrated ones at the end of simulations, which is 50 days for the OGCM and 10 days for the AGCM.  $I_S$  is the number of intermediate time steps between two GCM outputs. The "maximum time step" stands for the intermediate time step lengths ( $\Delta t_i$ ), which are used in the different trajectory integrations. The last column is the the computational time normalised with regard to the "time-dependent" case, where theoretical velocity fields are used to compute trajectories, i.e. with no data reading or writing.

|                | Distance to "time dependent" |      | $T_L$  | Maximum Time step    |              | Normalised    |
|----------------|------------------------------|------|--------|----------------------|--------------|---------------|
|                | OGCM                         | AGCM | AGCM   | OGCM                 | OGCM         | computational |
| $I_S$          | [km]                         | [km] | [days] | $\Delta t_i$         | $\Delta t_i$ | time          |
| "Fixed"        | 769                          | 4992 | 3.44   | $\equiv 5 \text{ d}$ | $\equiv 6 h$ | 0.830         |
| 1              | 276                          | 3835 | 3.88   | 5 d                  | 6 h          | 0.830         |
| 12             | 242                          | 2971 | 3.86   | 10 h                 | 30 min       | 2.110         |
| 120            | 103                          | 1752 | 3.86   | 1 h                  | 3 min        | 14.03         |
| 1,200          | 28                           | 1079 | 3.87   | 6 min                | 18 s         | 132.0         |
| 12,000         | 6                            | 1002 | 3.87   | 36 s                 | 2 s          | 1191          |
| Time dependent | 0                            | 0    | 3.87   | 5 d                  | 6 h          | 1.000         |

Figure 1. Schematic illustration of how the transport fields F(t) are updated and interpolated in time between the stored GCM data, which are read in at the time  $t^n$  and are separated in time by the time interval  $\Delta t_G$  (in red). The fields are then linearly interpolated at the blue points in blue with intermediate time steps. The number of intermediate time steps between two GCM velocities is in this example  $I_S = \Delta t_G / \Delta t_i = 4$ .

---

## Author Response (AR2)

Dear Bob,
We have made the changes you asked for.
Kristofer
PS. Good comments by the way.

- Page 2, line 7: "Their" should be "The" as first word

*This has been corrected*

- Page 2, line 13: This statement about reversibility of the flow not being possible in RK4 is somewhat mysterious. Are the authors here talking about a fundamental mathematical property of RK4? Because as far as I know the RK4 scheme is commutable under dt -> -dt. Or are they talking about numerical noise and accumulation of round-off errors? Because in that case the statement essentially says that the TRACMASS scheme has no (or far less)
numerical noise than RK4. I'd like to see evidence for that, if that's what the authors mean.

*We have removed ",which other trajectory methods, e.g. RK4, can not accomplish.".*

- Page 3, line 10: I think it would be good if the statement "since it does not satisfy the discretised continuity equation in a GCM" would be a bit more expanded. Why is this the case? Is this easy to derive? if so, please show, or otherwise provide references

*We have rephrased and softened this statement somewhat by writing on Page 3, Lines 8-12:*

*"An alternative approach is to assume that $u = u(x, y, z)$, $v = v(x, y, z)$ and $w = w(x, y, z)$ inside a grid cell, which might be more realistic in terms of representing unresolved motions. However, no such information is generally provided by GCMs. Furthermore, it would also require that the mass transports through the grid faces are unchanged in order to satisfy the continuity equation of the GCM."*

*and on Page 5, Lines 19-22, we added:*

*"Note that the calculation of the vertical mass transport $W_{i,j,k}^n$ through the top face of a grid box, with the Eqs. 12 – 15, only involves the velocities on the considered grid box. A 3D dependency of the velocities ($u = u(x, y, z)$, $v = v(x, y, z)$ and $w = w(x, y, z)$) would require velocities from other grid boxes, which could potentially break the mass conservation of Eqs. 12 – 15."*

- Figures 5, 6 and 7 would be easier to comprehend if they have latitude and

longitude ticks. While the Eurasian continent in Fig 7 is probably recognisable to most readers, I doubt whether the Agulhas region in Figures 5 and 6 is so recognisable too. Latitude and Longitude ticks will help readers orientate themselves.

*We have updated Figure 5 and 6 with longitudes and latitudes.*

---

## Author Response (AR3)

Dear Bob,

Embarassing that we missed to answer Stephen Griffies "specific comments", which were all good.

We have made all the changes he suggested, which are listed below.

Kristofer

**REPLY TO REFEREE GRIFFIES**

GENERAL COMMENTS

1. As detailed here, and in the earlier literature, the TRACMASS approach performs an analytic integration of the trajectory within a grid cell. This point is emphasized in the present manuscript. Importantly, this integration is enabled by an **assumption** that the subgrid scale velocity components are linear functions of their corresponding directions: [u(x), v(y), w(z)]. Surprisingly, this critical assumption is not explicity noted in the present manuscript. It should in fact be emphasized and defended.

How/where will it break down? As written, words such as "the trajectory solutions are exact" (pg 5, line 10) make it look like TRACMASS is performing magic. Instead, it is following an exact treatment based on the assumption of subgrid [u(x), v(y), w(z)].

*Answer: We have added a few sentences on this in the first paragraph of section 2 and section 2.2 to better highlight this key assumption. We agree that the use of the word "exact" may mislead readers, and have rewritten a few sentences such as p5, line10 to emphasise that the trajectories are solutions to a differential equation, and that there is nothing magical about it.*

2. The differential equations for the position within a grid cell are given by equation (17) for the stationary case, and equation (26) for the time-dependent case. Both equations are offered to the reader as if they should be an obvious consequence of something a priori. However, both equations need more build up to motivate and rationalize. The only statement to suggest where equation (17) comes from is line 19 on pg 5: "The transport and position within the grid box are now related by U = dr/dx...". However, this is a statement that offers no motivation nor a derivation. What is the basis for this relation?

So as written, equations (17) and (26) seemingly appear from no where, and the reader is left scratching his/her head. Sans shared intuition, these equations remain mere black boxes to the reader, which is of no use to the reader.

*Answer: We have rewritten the first paragraphs of section 2.2 (stationary scheme) and 2.4 (time-dependent scheme) to better lead up to the differential equations that we use to calculate trajectories. We hope this is clearer to the reader.*

3. At the end of Section 2, I found myself wanting to see a clear schematic to summarize the stationary method and the time-dependent method. Likely these schematics appear in

the basic literature. But given that you are rederiving the methods here, it would serve the reader well to have such schematics presented again, perhaps in an updated manner. These schematics could offer far more conceptual understanding than the maths presented in Section 2.

*Answer: A very good point and we have both added a paragraph at the end of section 2 and a figure, summarising the resulting differences between the schemes within a time-space cell.*

4. The word "this" is used many placed without qualifying. The reader is often left wondering what "this" referes to. Please be more careful with letting the reader know what "this" refers to. It is important to do so in order not to lose the reader, especially the novice.

*Answer: We have rewritten a number of sentences in order to remove "this".*

**SPECIFIC COMMENTS**

pg 2, line 25: "This in contrast..." suggest changing to "This method contrasts to the..."

*Answer: We have changed as you suggested.*

pg 3, line 11: change "an GCM" to "a GCM"

*Answer: We have changed as you suggested.*

pg 4, equation (4): The grid cell thickness is a function of (i,j,k,n). However, this space-time dependence is not consistently displayed in the manuscript, such as in equation (8) where we only see dependence on (k,n). Where it is relevant, and where (i,j) are exhibited for Delta x and Delta y, please also display such dependence for Delta z.

*Answer: We have changed this in Eqs 11, 12, 13, 5, 16 as well as lines 27 and 28.*

pg 4, line 17: not all ocean GCMs are incompressible (e.g., MOM and MITgcm have non-Boussinesq compressible options). So please qualify this statement; e.g., "many ocean models are incompressible".

*Answer: We have rephrased this to "Note that the mass transport can be replaced by the volume transport in models that assume the fluid to be incompressible, which is the case for most OGCMs."*

pg 4, equations (9), (10), (12): As written, these equations are not usable when integrating within the model code online, unless we have the n+1 value of the thickness, density, and/or pressure. So please comment on what operationally you mean by these equations.

*Answer: We have changed the finite differences in Eqs 12, 13 and 15 to backward-difference schemes and added the sentence: "Note that in the case of "off-line" calculations, one may instead use centred or forward finite time differences in Eqs. 12, 13 and 15."*

pg 5, line 16: It is useful to here state that r = x/Delta x is defined separately for each direction and for each grid cell. Namely, there is no assumption that Delta x is uniform across the model.

*Answer: We have now instead included the index i,j to Delta x and added the sentence at line 12 p.6: " The above procedure is repeated for meridional and vertical displacements, where now ..."*

pg 5, line 16: change "velocity" to "transport"

*Answer: We have rewritten the beginning of section 2.2. Starting with velocities and then replacing it with volume or mass flux.*

pg 6, lines 9-10: this sentence appears to be an orphan from another paragraph. Perhaps it should go into the previous paragraph, though with some editing to reduce redundancy.

*Answer: It was in the wrong position and has been removed now.*

pg 6, line 30: remove "with" at start of line.

*Answer: We removed "with".*

pg 6, line 30: Global GCMs are often run with time steps of **hours** to minutes. For example, the GFDL-CM2.1 ocean component, which is one degree, uses a 2hours time step. GFDL-CM2.5, which is 1/4 degree, uses 30minutes.

*Answer: We have now written: "If this is undertaken "on-line", i.e., in the same time as the GCM is integrated, this time interval will simply be the same as the time step the GCM is integrated, which is typically between minutes to a few hours in a global GCM."*

pg 7, line 11: I believe that "spatial resolution" should be changed to "spatial grid spacing". Namely, resolution is a pure number whereas grid spacing has units of length. You are stating (Delta x,Delta y,Delta z), which is grid spacing, not resolution.

*Answer: We have changed it as you suggested.*

pg 7, equation (25): which time level do you use for Delta z in the case where the grid cell thickness is a function of time? n? n-1? average?

*Answer: We have included at page 8, line 7, the sentence: "The vertical grid box spacing is for models with time dependent grid cell thicknesses replaced with an average between the two time steps ...".*

pg 9, line 15: suggest "...is that in the time-dependent scheme, the transit times..."

*Answer: Changed as suggested.*

pg 11, line 8 and title to Section 3.1: No one has proven that 1/12th degree **resolves** mesoscale eddies globally. Instead, such models **admit** mesoscale eddies, but they are surely not fully resolved globally. So to state the ORCA12 is "eddy resolving" is an over-statement that is not justified.

*Answer: Yes, you are right and we have replaced "eddy resolving" with "high resolution".*

pg 12, line 32: what is "this" referring to in the middle of the line? pg 13, line 17: "the the"

*Answer: This section has now been rewritten.*

pg 13, line 32: "...quantify this **difference** we compared..."

*Answer: Changed as you suggested.*

pg 14, line 26: It is stated here that the time-dependent scheme is more accurate than the alternatives. I suspect that it is indeed more accurate. But I see no where in the paper where a "truth" is used to base this conclusion. Or did I miss something??

*Answer: We havre rephrased and softened this by writing: "The study has shown that the TRACMASS "time-dependent" scheme is likely to be more accurate as well as faster than the "stepwise-stationary" scheme with intermediate steps."*

pg 15, lines 14, 16, 25: please specify what "this" refers to on each of these lines. The reader should not be asked to assume he/she knows what you are referring to.

*Answer: We have now rewritten the three sentences: 1) "The mass transport was tested in the Agulhas experiment, where the "fixed GCM time step" scheme favoured relatively the Agulhas retroflection with more trajectories returning into the Indian compared to the*

*"time-dependent" and "stepwise-stationary" schemes., 2) "This difference in mass transport can be explained by the delicate path of the Agulhas leakage, which requires an accurate temporal evolution so that particles can be retained in Agulhas rings.", 3) "This requirement of mass conservation will always be somewhat more demanding than for other trajectory codes, since it requires a total understanding of the various GCM coordinate systems as well as incorporating them in the TRACMASS framework."*

Figure 1 caption. Remove first "blue" in the second sentence. C4

*Answer: Removed.*

Figure 2 caption "the the"

*Answer: Corrected.*